# FedPolicy: An RL-Guided Redistribution Policy for Synergizing Local-Global Optimization in Federated Learning

## Abstract

Statistical heterogeneity remains a central challenge in federated learning. Existing methods primarily address this problem through improved local objectives, aggregation strategies, or personalization mechanisms, while the post-aggregation redistribution step is typically applied uniformly and has received comparatively little explicit attention. This design becomes problematic under heterogeneous client distributions, where repeatedly overwriting local models with the same aggregated parameters can disrupt client-specific adaptation and induce negative transfer. We propose FedPolicy, an RL-guided post-aggregation redistribution framework that treats the return path of the aggregated model as a client-specific decision problem. Rather than broadcasting the same update to every client, FedPolicy learns which part of the aggregated model should be transferred back to each client by selecting among full-model, backbone-only, and head-only parameter blocks. This formulation identifies post-aggregation redistribution as a previously underexplored control axis in federated optimization, improving the balance between global transfer and local specialization. Extensive experiments under heterogeneous federated settings show that FedPolicy consistently outperforms strong baselines across FMNIST, CIFAR-10, CIFAR-100, and ISIC2019, with the clearest gains appearing in the more challenging heterogeneous regimes. Across all four datasets and heterogeneity settings, FedPolicy achieves an average relative gain of approximately 5.85% over the strongest baseline, with the largest relative improvement reaching 16.47% on ISIC2019 under severe heterogeneity, while converging faster and delivering a more favorable cost-to-accuracy trade-off with negligible overhead. These results highlight client-specific post-aggregation redistribution as an underexplored yet impactful design dimension in heterogeneous federated learning. Code for reproducing the results is available at `https://anonymous.4open.science/r/A-FedPolicy-816B`.

## 1 Introduction

Federated learning (FL) systems commonly assume that once a global model is obtained through aggregation, it can be applied *uniformly* to all participating clients without adverse effects. However, in practice, clients often differ substantially in data distribution, representation quality, and class-level learning difficulty. Under such heterogeneity, directly overwriting local models with aggregated parameters uniformly can induce negative transfer, destabilize training dynamics, and degrade the effectiveness of subsequent aggregation rounds. Despite substantial progress in federated learning, the question of *how aggregated parameters should be applied to heterogeneous clients* remains underexplored.

Intuitively, clients that have already learned stable and specialized representations may suffer representation erosion when aggressively aligned with the global model, whereas clients struggling with specific classes may propagate noisy or biased updates if forced to fully absorb global parameters. The effect is not limited to a single round. Post-aggregation initialization determines which local trajectory each client follows before returning updates to the server. If initialization is mismatched, the returned updates may become noisier or less aligned with local data geometry, which then degrades the next aggregate and propagates instability across rounds. By contrast, client-specific redistribution can preserve useful local structure while still exploiting transferable global progress. Addressing this mismatch becomes crucial, especially under severe

non-independent and identically distributed (non-IID) data distributions, as client updates directly influence the quality and stability of future aggregation, yet existing FL pipelines treat this step as a fixed and uniform operation.

In the standard federated learning framework, clients perform local optimization, the server aggregates their updates, and the resulting global model is redistributed to all clients for the next round McMahan et al. (2017). Most prior federated learning methods address heterogeneity through two main design axes: *local objective design* and *server-side aggregation design*. Regularization-based approaches such as FedProx Li et al. (2020) and control-variate methods such as SCAFFOLD Karimireddy et al. (2020) constrain local optimization to mitigate client drift. Adaptive aggregation strategies Reddi et al. (2021); Wang et al. (2020b); Chen et al. (2023) seek to combine heterogeneous updates more robustly at the server, while client-adaptive methods Arivazhagan et al. (2019); Wang et al. (2024b); Grinwald et al. (2024) and prototype- or representation-learning approaches Tan et al. (2022); Collins et al. (2021); Yang et al. (2024) further tailor learned representations to client-specific distributions. More recently, our prior work FedCA Chowdhury & Halder (2026) extended this direction by jointly addressing heterogeneity through confusion-aware local optimization and class-prioritized aggregation under extreme non-IID settings. Readers may refer to Appendix A for more details.

However, a common assumption still remains unchanged across these lines of work: once the global model is formed, it is redistributed back to all participating clients in the same form. Although prior work has substantially improved both *how clients are trained locally* and *how their updates are aggregated globally*, it still leaves underexplored *how the aggregated global model should be transferred back to heterogeneous clients*. This gap is especially consequential under strong heterogeneity, where the same global update need not be equally beneficial to all clients. A client may benefit from global representation transfer while needing to preserve its local classifier, whereas another may require boundary correction without disturbing already adapted features. Treating redistribution as uniformly beneficial therefore introduces a mismatch between the aggregated model and the client's local learning state, which can degrade both local adaptation and the quality of subsequent aggregation rounds.

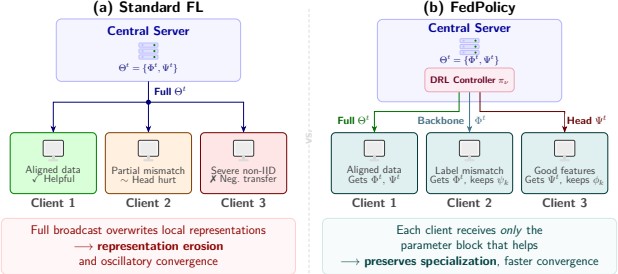

Figure 1: **Selective parameter broadcast under heterogeneity.** **(a)** Uniform full-model broadcast can induce representation drift and negative transfer when client data are non-IID. **(b)** FedPolicy uses a DRL policy to choose a client-specific broadcast action among full model ($\Theta^t$), backbone-only ($\Phi^t$), and head-only ($\Psi^t$). This reflects the backbone/head functional separation and selectively preserves beneficial local structure while leveraging global progress.

In this work, we argue that effective federated learning under heterogeneity requires not only better local objectives or better aggregation operators, but also explicit control over *how the aggregated global model is transferred back to individual clients*. This post-aggregation step is inherently sequential and non-stationary: each initialization decision changes a client's local training trajectory, and the resulting local updates affect the aggregate model used in the subsequent round. Moreover, because the participating clients, class composition, and model states vary across rounds, the learning environment is itself non-stationary. As a result, fixed heuristic rules may not remain effective throughout training. These characteristics naturally motivate a reinforcement learning (RL) formulation.

To this end, we propose FedPolicy, an RL policy-guided federated learning framework that treats post-aggregation redistribution as an adaptive, client-specific decision problem. After the server forms the global model at a communication round, FedPolicy selects whether each participating client should receive the full global model, only the backbone, or only the classifier head, while leaving local optimization and server-side aggregation unchanged. This selective-initialization perspective is illustrated in Fig. 1. The overall contributions are summarized as follows:

- We introduce FedPolicy, a **Policy-Guided Selective Global Redistribution** framework that formulates client-specific post-aggregation redistribution as a sequential decision-making problem and adaptively selects Full, Backbone, or Head redistribution for each selected client.

- We design a composite client state representation that captures **semantic error topology** via soft confusion matrices and **optimization status** via parameter divergence and validation performance gaps, providing a compact signal for heterogeneity-aware redistribution control.

- We provide a limited optimization characterization of selective initialization, clarifying when a specified block-wise redistribution action can improve the immediate client objective and how realized initialization mismatch enters a stationarity bound.

- We conduct extensive experiments on FMNIST, CIFAR-10, CIFAR-100, and ISIC2019 under heterogeneous federated settings, comparing against standard, personalized, confusion-aware, and partial-sharing baselines, together with controller, state, reward, dropout, and aggregation-compatibility ablations.

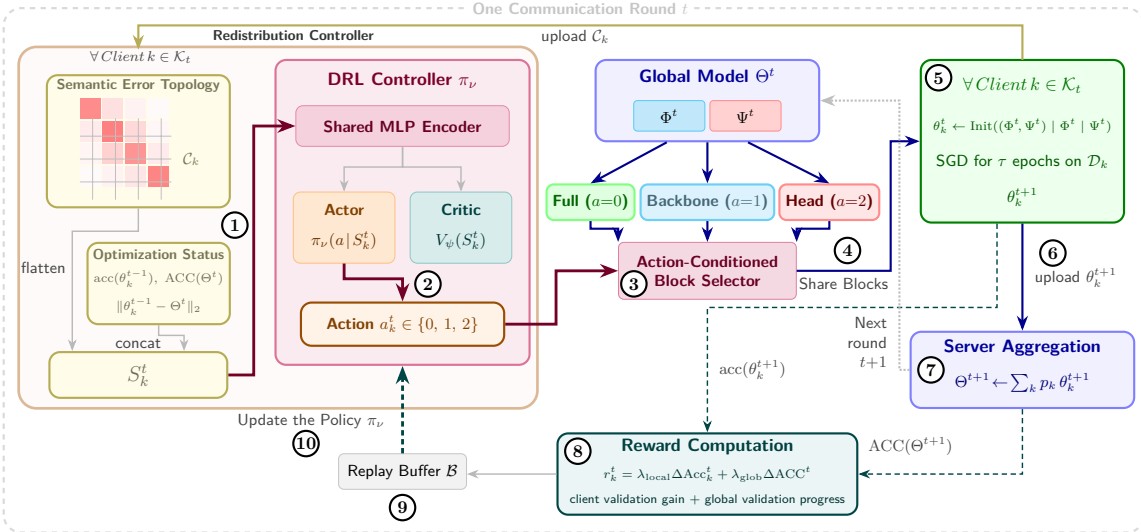

Figure 2: Overview of the FedPolicy training workflow.

## 2 Methodology

This section presents FedPolicy, a policy-guided federated learning framework for client-specific post-aggregation parameter sharing under statistical heterogeneity. Figure 2 illustrates the workflow of a training round. The redistribution controller is the key component of FedPolicy, which allows the server to construct a client-specific state representation for each selected client and obtain a sharing action, that is, whether to share the full model, the backbone only, or the head only (Steps 1–2). Based on this action, the server determines and transmits the corresponding parameter block (Steps 3–4). Following this, the client performs local training by incorporating the received block into its local model, and returns the updated model together with the confusion statistics required for policy evaluation (Steps 5–6). Notably, FedPolicy does not introduce adaptivity at the aggregation stage. The server forms the global model using the standard mean aggregation rule, and measures global progress on a small server-held-out validation split, which is then combined with client-side validation improvement to assess the effect of the selected redistribution actions and generate the reward signal for training the redistribution controller (Steps 7–10). In this way, FedPolicy concentrates adaptivity in the post-aggregation parameter redistribution stage while retaining standard local optimization and server aggregation. Table 1 summarizes the key mathematical symbols and notation used throughout the paper.

Table 1: Key notations and symbols.

| Symbol | Description |
|---|---|
| $\mathcal{A}$ | Action space of the controller |
| $a_k^t$ | Action selected for client $k$ at round $t$ |
| $\beta$ | EMA momentum for soft confusion tracking |
| $C_k$ | Soft confusion matrix of client $k$ |
| $\mathcal{B}$ | Replay buffer for DRL training |
| $\mathcal{D}_k, \mathcal{D}_k^{\mathrm{val}}$ | Local training and validation datasets of client $k$ |
| $\mathrm{acc}(\cdot)$ | Client-side validation accuracy |
| $\mathrm{ACC}(\cdot)$ | Server-side held-out validation accuracy |
| $\Delta\mathrm{Acc}_k^t$ | Local validation improvement of client $k$ |
| $\Delta\mathrm{ACC}^t$ | Server-side held-out validation improvement after round $t$ |
| $\mathrm{vec}(\cdot)$ | Vectorization operator |
| $\eta$ | Local learning rate |
| $\gamma$ | DRL discount factor |
| $\mathcal{J}(\Theta)$ | Global objective |
| $\mathcal{K}_t$ | Set of participating clients at round $t$ |
| $\lambda_{\mathrm{local}}$ | Weight of the local reward term |
| $\lambda_{\mathrm{glob}}$ | Weight of the global reward term |
| $\mathcal{L}_k$ | Empirical risk on client $k$ |
| $\Phi$ | Backbone (representation encoder) parameters |
| $\pi_\nu$ | DRL policy with parameters $\nu$ |
| $p_k$ | Aggregation weight for client $k$ |
| $r_k^t$ | Reward of client $k$ at round $t$ |
| $\mathcal{S}$ | State space |
| $S_k^t$ | State of client $k$ at round $t$ |
| $\tau$ | Number of local training epochs |
| $\Theta^t = \{\Phi^t, \Psi^t\}$ | Global model at round $t$ |
| $\theta_k^t = \{\phi_k^t, \psi_k^t\}$ | Local model of client $k$ at round $t$ |
| $\Psi$ | Classifier head parameters |

## 2.1 Problem Formulation

We first recall the standard federated learning objective. Given $N$ clients with local empirical risks $\mathcal{L}_k$ and aggregation weights $p_k$, conventional FL seeks a single global model

$$\min_{\Theta}\ \mathcal{J}(\Theta) = \min_{\Theta} \sum_{k=1}^{N} p_k \mathcal{L}_k(\Theta), \tag{1}$$

and then broadcasts the same aggregated model to all selected clients at the next round. FedPolicy keeps this local-training and aggregation pipeline fixed, but replaces the uniform post-aggregation broadcast with a client-specific redistribution decision.

We decompose the global model parameters $\Theta$ into two functional subspaces: the representation encoder (backbone) $\Phi$ and the classifier head $\Psi$, i.e., $\Theta = \{\Phi, \Psi\}$. This decomposition reflects the observation that statistical heterogeneity affects representation learning and decision boundaries asymmetrically. The backbone primarily captures transferable features, whereas the classifier head is more sensitive to local label composition.

The federated optimization process proceeds in communication rounds $t = 1, \ldots, T$. At round $t$, the server maintains the aggregated model $\Theta^t = \{\Phi^t, \Psi^t\}$ and selects a subset of clients $\mathcal{K}_t$ for participation. For each selected client $k \in \mathcal{K}_t$, the server chooses an action $a_k^t$ that determines which parameter block of $\Theta^t$ should

be shared with that client. We denote the transmitted block by

$$\zeta_k^t = \text{Share}(\Theta^t, a_k^t), \tag{2}$$

where $\zeta_k^t$ may correspond to the *full model*, the *backbone* only, or the *head* only. The client then combines the received block with its previous local model $\theta_k^{t-1}$ to form the effective model $\theta_k^t$ used for local training.

Given this selectively initialized model, client $k$ performs local optimization on its private dataset $\mathcal{D}_k$ for $\tau$ epochs to obtain an updated model $\theta_k^{t+1}$:

$$\theta_k^{t+1} \leftarrow \theta_k^t - \eta \nabla \mathcal{L}_k(\theta_k^t), \tag{3}$$

where $\eta$ is the local learning rate and $\mathcal{L}_k$ denotes the empirical risk on client $k$.

After receiving the updated local models from the participating clients, the server forms the next global model through standard aggregation:

$$\Theta^{t+1} \leftarrow \sum_{k \in \mathcal{K}_t} p_k \theta_k^{t+1}, \tag{4}$$

where client's contribution weight $p_k = 1/|\mathcal{K}_t|$ for all $k \in \mathcal{K}_t$, for mean (FedAvg-style) aggregation.

Our objective is to learn a policy $\pi_\nu$ that selects actions $a_k^t$ so as to improve both client-level adaptation and the quality of future global aggregation over the communication horizon. Formally, we seek

$$\min_{\pi_\nu} \mathbb{E}\big[\mathcal{J}(\Theta^T)\big] = \min_{\pi_\nu} \mathbb{E}\left[\frac{1}{N} \sum_{k=1}^N \mathcal{L}_k(\Theta^T)\right], \tag{5}$$

where $N$ denotes the total number of clients in the federation. This formulation elevates post-aggregation parameter sharing from a fixed implementation step to a sequential control problem in federated learning.

## 2.2 Fixed Local Training and Aggregation Pipeline

FedPolicy preserves the standard client–server optimization loop: participating clients perform local empirical risk minimization, and the server aggregates the resulting updates into the next global model. The intervention of the proposed method occurs only after this aggregation step, when the aggregated model is redistributed to clients for the next round of local training. For a selected client $k$, the local objective is,

$$\mathcal{L}_k(\theta_k^t) = \frac{1}{|\mathcal{D}_k|} \sum_{(x,y) \in \mathcal{D}_k} \ell\Big(f_{\theta_k^t}(x), y\Big), \tag{6}$$

where $\ell(\cdot, \cdot)$ denotes the task loss and $f_{\theta_k^t}$ is the local model obtained after selective parameter sharing. After $\tau$ local epochs, each client returns the updated model $\theta_k^{t+1}$, the corresponding soft confusion statistics $C_k$, and the validation accuracy signal $\text{acc}(\theta_k^{t+1})$ used for policy learning.

The global model is then updated using the standard mean aggregation rule in Eq. (4). This design choice is deliberate as prior heterogeneity-aware studies including FedCA, improved robustness by adapting local objectives and aggregation-side coordination. In contrast, FedPolicy targets a different unresolved stage of the FL pipeline: once an aggregated model has already been formed, how should it be redistributed back to heterogeneous clients? By keeping aggregation fixed, the method isolates redistribution as the control point and attributes gains to client-specific parameter sharing rather than aggregation complexity.

## 2.3 RL-Guided Post-Aggregation Redistribution Controller

We cast post-aggregation redistribution as a sequential decision problem executed at the server. Unlike prior RL-based FL methods that typically learn policies for client selection, aggregation weighting, resource allocation, or hyperparameter control, FedPolicy applies the RL controller after standard aggregation to choose which part of the already aggregated model should be redistributed to each selected client. Thus,

the novelty is not the use of RL alone, but the formulation of post-aggregation redistribution as a client-conditioned sequential control problem. At communication round $t$, once the aggregated model $\Theta^t$ has been obtained, the server must decide, for each selected client $k \in \mathcal{K}_t$, which parameter block of $\Theta^t$ should be shared with that client, denoted as $a_k^t \in \mathcal{A}$. The action $a_k^t$ determines the shared block $\zeta_k^t$ and, through client-side initialization, the effective local model $\theta_k^t$ used for the next local optimization phase. However, a central challenge is that the quality of $a_k^t$ cannot be reliably judged from immediate observations alone. The selected sharing action influences both the client's short-term local training trajectory and the quality of the returned update $\theta_k^{t+1}$, which in turn shapes the subsequent aggregate $\Theta^{t+1}$. The effect of each decision therefore propagates through coupled client-level and federation-level dynamics, making post-aggregation redistribution a sequential control problem rather than a static assignment rule. Accordingly, we model the controller as a DRL policy

$$\pi_\nu(a_k^t \mid S_k^t), \tag{7}$$

where $S_k^t$ is a client-specific state representation and $\nu$ denotes the controller parameters. For each selected client, the server observes $S_k^t$, selects an action $a_k^t$, transmits the corresponding block $\zeta_k^t$, and receives feedback only after local optimization and subsequent aggregation. The resulting interaction induces a finite-horizon Markov decision process with transitions of the form $(S_k^t, a_k^t, r_k^t, S_k^{t+1})$, from which the controller is trained.

The objective of the controller is to maximize the expected long-term utility of client-specific redistribution over the communication horizon:

$$\max_\nu \ \mathbb{E}_{\pi_\nu} \left[ \sum_{t=0}^{T-1} \gamma^t r_k^t \right], \tag{8}$$

where $\gamma \in [0,1]$ is the discount factor.

## 2.4 State Representation: Semantic and Optimization Signals

The controller requires a client-specific state $S_k^t$ that captures both the nature of the client's learning difficulty and its relationship to the current global model. In FedPolicy, the state is designed to encode two complementary aspects of heterogeneity: semantic error structure and optimization status.

### 2.4.1 Semantic Error Structure

A scalar metric such as total loss or top-1 accuracy cannot reveal which classes a client consistently confuses. Yet such class-level structure is important for determining whether the client should preserve its local classifier, adopt the global classifier, or absorb the full global model. To represent this information compactly, we use a soft confusion matrix $C_k \in \mathbb{R}^{C \times C}$, where $C$ is the number of classes and each entry reflects the tendency of the current local model to predict class $j$ for a sample whose true label is class $i$. The flattened representation $\mathrm{vec}(C_k)$ therefore serves as a compact summary of the client's class-level confusion structure.

### 2.4.2 Optimization Status

In addition to semantic confusion, the controller must assess how the client's local model relates to the current global model. For this purpose, we include two optimization-status signals.

First, we use the client-side validation accuracy $\mathrm{acc}(\theta_k^{t-1})$ from the previous round together with the current server-side held-out validation accuracy $\mathrm{ACC}(\Theta^t)$. These quantities indicate whether the incoming global model is likely to be beneficial relative to the client's current local state.

Second, we measure the parameter-space mismatch between the local and global models as $d_k^t = \left\| \theta_k^{t-1} - \Theta^t \right\|_2$. A large mismatch indicates that the client has evolved along a trajectory substantially different from the global consensus. We use this quantity as a state signal for the controller, not as evidence that selective redistribution is always preferable to a full-model overwrite.

Combining both semantic and optimization signals, the state vector for client $k$ at round $t$ is defined as

$$S_k^t = \left[ \mathrm{vec}(C_k) \oplus \mathrm{acc}(\theta_k^{t-1}) \oplus \mathrm{ACC}(\Theta^t) \oplus d_k^t \right], \tag{9}$$

where $\oplus$ denotes vector concatenation.

## 2.5 Parameter Block Selection

Given the client state $S_k^t$, the controller selects an action $a_k^t \in \mathcal{A} = \{0, 1, 2\}$, which specifies the parameter-sharing rule and the transmitted global block $\zeta_k^t$, thereby determining the effective local model $\theta_k^t$.

Concretely, (i) for $a_k^t = 0$, the server broadcasts the full aggregated model $\zeta_k^t = \{\Phi^t, \Psi^t\}$, yielding $\theta_k^t = \{\Phi^t, \Psi^t\}$ and recovering standard federated learning; this is suitable when the client is well aligned with the global model. (ii) For $a_k^t = 1$, only the global backbone is transmitted, $\zeta_k^t = \Phi^t$, while the client retains its local head, resulting in $\theta_k^t = \{\Phi^t, \psi_k^{t-1}\}$; this preserves client-specific decision boundaries while sharing feature representations. (iii) For $a_k^t = 2$, only the global head is transmitted, $\zeta_k^t = \Psi^t$, yielding $\theta_k^t = \{\phi_k^{t-1}, \Psi^t\}$; this preserves locally adapted features while aligning the classifier with the global decision structure. FedPolicy uses a deliberately compact action space with three redistribution choices. These actions preserve a simple controller design while capturing three meaningful levels of global knowledge transfer: full-model redistribution, backbone-level representation transfer, and classifier-head alignment. The selected action only controls the post-aggregation

---

**Algorithm 1**: Federated Learning with Policy-Guided Selective Parameter Sharing (FedPolicy)

---

**Require:** Global model $\Theta^t$, previous client models $\{\theta_k^{t-1}\}$, controller policy $\pi_\nu$, replay buffer $\mathcal{B}$
**Ensure:** Next global model $\Theta^{t+1}$, updated controller parameters $\nu$

**// Phase 1: State construction and policy-guided sharing**
1: **for** each selected client $k \in \mathcal{K}_t$ **in parallel do**
2:     Construct state $S_k^t$ using $\theta_k^{t-1}$ and $\Theta^t$ (Eq. 9)
3:     Select action $a_k^t \sim \pi_\nu(\cdot \mid S_k^t)$
4:     Determine shared block $\zeta_k^t \leftarrow \text{Share}(\Theta^t, a_k^t)$
5:     Initialize $\theta_k^t$ from $\theta_k^{t-1}$ using $\zeta_k^t$
6: **end for**
**// Phase 2: Local training**
7: **for** each selected client $k \in \mathcal{K}_t$ **in parallel do**
8:     Update $\theta_k^{t+1}$ via local training on $\mathcal{D}_k$
9:     Return $\theta_k^{t+1}$ and client-side feedback to the server
10: **end for**
**// Phase 3: Server aggregation**
11: Aggregate returned models using Eq. (4)
12: **for** each selected client $k \in \mathcal{K}_t$ **do**
13:     Compute reward $r_k^t$ using Eq. (10) and next state $S_k^{t+1}$
14:     Store transition $(S_k^t, a_k^t, r_k^t, S_k^{t+1})$ in $\mathcal{B}$
15: **end for**
**// Phase 4: Controller update**
16: **if** update interval reached **then**
17:     Sample a minibatch from $\mathcal{B}$
18:     Update controller parameters $\nu$ using the chosen DRL objective
19: **end if**

---

redistribution step; it does not change which layers are locally trained or how client updates are aggregated. Unlike FedPart, where participating clients follow a predefined layer-wise schedule for training and aggregation, FedPolicy first performs standard local training and aggregation, and then learns a client- and round-specific decision about whether the Full model, Backbone, or Head should be redistributed. Thus, the contribution is not partial parameter exchange alone, but learned control of the post-aggregation transfer step. This also distinguishes FedPolicy from personalized FL methods that maintain client-specific models or predefined shared/private parameter partitions. FedPolicy does not introduce a permanent personalized component or a separate personalization objective; the same client may receive different redistribution actions across rounds as its state evolves.

## 2.6 Policy Learning and Reward Design

We define the controller problem as a finite-horizon Markov decision process $(\mathcal{S}, \mathcal{A}, \mathcal{P}, \mathcal{R}, \gamma)$, where $\mathcal{S}$ is the state space defined by Eq. (9), $\mathcal{A}$ is the action space of parameter-block broadcast, $\mathcal{P}$ captures the stochastic transition dynamics induced by local training and server-side aggregation, $\mathcal{R}$ is the reward function, and $\gamma \in (0, 1)$ is the discount factor.

To guide learning, we define a reward that combines client-level improvement with federation-level progress:

$$r_k^t = \lambda_{\text{local}} \Delta \text{Acc}_k^t + \lambda_{\text{glob}} \Big( \text{ACC}(\Theta^{t+1}) - \text{ACC}(\Theta^t) \Big), \tag{10}$$

where $\Delta \text{Acc}_k^t = \text{acc}(\theta_k^{t+1}) - \text{acc}(\theta_k^{t-1})$, $\Delta \text{ACC}^t = \text{ACC}(\Theta^{t+1}) - \text{ACC}(\Theta^t)$, and $\lambda_{\text{local}}$ and $\lambda_{\text{glob}}$ control the local and federation-level reward contributions, respectively. Here, $\theta_k^{t-1}$ refers to the client's previous local model prior to the round-$t$ broadcast, whereas $\theta_k^{t+1}$ is the model obtained after incorporating the selected parameter block and completing local training. Thus, $\Delta \text{Acc}_k^t$ measures the client-side improvement produced over the round-$t$ transition. The first term rewards local improvement induced by the selected broadcast action, while the second term provides a shared federation-level signal that reflects whether the resulting

round contributes positively to global validation progress. Appendix C.4.2 reports a sensitivity study over both local and global reward weights. Consequently, a redistribution action that improves an individual client's local state but degrades the subsequently aggregated global model receives a reduced reward through the global-progress term. This does not constitute a theoretical guarantee that the controller always avoids destabilizing actions, but it aligns policy learning with the quality of the post-aggregation model rather than only with local adaptation. The test set is reserved strictly for final reporting and is never used for controller training, reward computation, state construction, or policy updates.

The controller is trained to maximize the expected discounted return in Eq. (8). Since the DRL module operates entirely at the server, clients do not perform any policy-learning computation locally; they only receive the selected parameter block and then follow the standard federated training routine.

Overall, FedPolicy provides an end-to-end, policy-guided mechanism for client-specific post-aggregation parameter transfer, enabling the federation to balance global knowledge sharing with preservation of locally specialized structure under heterogeneous data distributions. Algorithm 1 summarizes the complete training workflow. For each selected client, the server constructs a client state, selects a broadcast action, determines the corresponding parameter block, and forms the effective local model. The selected clients then perform local training and return their updated models together with the statistics required for policy evaluation. The server first aggregates the returned client models using the fixed mean/FedAvg-style rule, and then computes reward and next-state information, stores the resulting transition in the replay buffer, and updates the controller from replayed transitions. This cycle is repeated at each communication round until convergence or until the communication budget is exhausted.

## 3 Theoretical Analysis

We study the optimization effect of selective post-aggregation initialization for a specified redistribution action. The analysis first gives a local client-level comparison showing when Backbone or Head redistribution can yield a lower immediate objective than Full overwrite, and then derives a one-step full-participation stationarity bound showing how the realized selective-initialization mismatch enters the optimization error. This characterization isolates the effect of selective initialization after an action has been chosen. It does not analyze the learning dynamics of the RL controller, guarantee that the controller selects an improving action, or establish that FedPolicy has lower mismatch than uniform redistribution. Full derivations are deferred to Appendix B.

**Assumption 1** (Smoothness). *For each client $k$, the local objective $\mathcal{L}_k$ is $L$-smooth. Equivalently, for all $u, v \in \mathbb{R}^d$,*

$$\mathcal{L}_k(u) \leq \mathcal{L}_k(v) + \langle \nabla \mathcal{L}_k(v), u - v \rangle + \frac{L}{2}\|u - v\|^2.$$

**Assumption 2** (Block-wise Strong Convexity). *In a neighborhood of candidate updates, $\mathcal{L}_k([\phi; \psi])$ is $\mu$-strongly convex in each block when the other block is fixed.*

**Assumption 3** (Drift-Bounded Block Gradient). *For the selected client state, the candidate partially updated models satisfy*

$$\|\nabla_\psi \mathcal{L}_k(\mathbf{w}^t_{Back})\| \leq \epsilon_{\mathrm{drift}}, \qquad \|\nabla_\phi \mathcal{L}_k(\mathbf{w}^t_{Head})\| \leq \epsilon_{\mathrm{drift}}.$$

**Theorem 1** (Block-wise Redistribution Comparison Bound). *Under Assumptions 2 and 3, the Backbone and Head redistribution candidates satisfy*

$$\mathcal{L}_k(\mathbf{w}^t_{Full}) - \mathcal{L}_k(\mathbf{w}^t_{Back}) \geq \frac{\mu}{4}\|\psi^t - \psi_k^{t-1}\|^2 - \frac{\epsilon_{\mathrm{drift}}^2}{\mu}, \tag{11}$$

$$\mathcal{L}_k(\mathbf{w}^t_{Full}) - \mathcal{L}_k(\mathbf{w}^t_{Head}) \geq \frac{\mu}{4}\|\phi^t - \phi_k^{t-1}\|^2 - \frac{\epsilon_{\mathrm{drift}}^2}{\mu}. \tag{12}$$

*Consequently, Backbone redistribution has strictly lower client loss than Full overwrite whenever $\|\psi^t - \psi_k^{t-1}\| > 2\epsilon_{\mathrm{drift}}/\mu$, and Head redistribution has strictly lower client loss whenever $\|\phi^t - \phi_k^{t-1}\| > 2\epsilon_{\mathrm{drift}}/\mu$.*

**Theorem 2** (Stationarity Characterization under Selective-Initialization Error)**.** *Under Assumption 1, let* $f(\mathbf{w}) = \sum_{k=1}^{N} p_k \mathcal{L}_k(\mathbf{w})$, *where* $p_k \geq 0$ *and* $\sum_k p_k = 1$. *Consider the full-participation, one-exact-local-step abstraction*

$$\tilde{\mathbf{w}}_k^t = \mathbf{w}^t + \boldsymbol{\delta}_k^t, \qquad \mathbf{w}^{t+1} = \sum_{k=1}^{N} p_k \left( \tilde{\mathbf{w}}_k^t - \eta \nabla \mathcal{L}_k(\tilde{\mathbf{w}}_k^t) \right), \tag{13}$$

*with* $0 < \eta \leq 1/L$. *Define* $D_t^2 := \sum_k p_k \|\boldsymbol{\delta}_k^t\|^2$ *and* $\bar{\boldsymbol{\delta}}^t := \sum_k p_k \boldsymbol{\delta}_k^t$. *If* $f$ *is bounded below by* $f^\star$, *then*

$$\frac{1}{T} \sum_{t=0}^{T-1} \|\nabla f(\mathbf{w}^t)\|^2 \leq \frac{2(f(\mathbf{w}^0) - f^\star)}{\eta T} + \frac{2L^2}{T} \sum_{t=0}^{T-1} D_t^2 + \frac{2}{\eta^2 T} \sum_{t=0}^{T-1} \|\bar{\boldsymbol{\delta}}^t\|^2. \tag{14}$$

*Thus the average stationarity measure converges to zero when the two time-averaged mismatch terms on the right vanish; otherwise the theorem characterizes the corresponding error neighborhood. For Full overwrite,* $\boldsymbol{\delta}_k^t = 0$ *and both mismatch terms disappear. The characterization therefore makes the dependence on the realized initialization offsets explicit; when these offsets are nonzero, they determine the size of the stationarity neighborhood.*

Together, Theorems 1 and 2 provide a limited optimization view of selective initialization after a redistribution action has been chosen. The block-wise comparison identifies local conditions under which retaining one parameter block while updating the other can reduce the immediate client objective relative to Full overwrite. The stationarity characterization complements this local view by expressing selective initialization as an exact inexact-gradient update, where the realized initialization offsets enter the global bound through $D_t^2$ and $\|\bar{\boldsymbol{\delta}}^t\|^2$. The analysis therefore distinguishes the possible local adaptation benefit from the global stability cost: parameter divergence serves as a controller state signal, and the reward signal determines empirically whether the chosen redistribution improves the subsequently aggregated model.

It is worth noting that the role of the learned controller remains empirical. Neither Theorem 1 nor Theorem 2 assumes that the controller selects the action with the smallest mismatch. Consequently, these theorems do not establish policy optimality, systematic mismatch reduction, or an ordering between FedPolicy and Full redistribution. We evaluate learned action selection empirically against fixed Full-Only, Backbone-Only, Head-Only, and Random strategies, as well as a contextual bandit and multiple RL controllers. Extending the analysis to partial participation, multiple stochastic local steps, and coupled policy learning would require additional sampling, variance, and controller assumptions, and is left for future work.

## 4 Experimental Evaluation

We evaluate FedPolicy under heterogeneous client distributions generated by both probabilistic label-skew and deterministic class-absence partitions. The study is designed to answer four questions: (i) whether client-specific post-aggregation redistribution improves accuracy under heterogeneity, (ii) whether it accelerates convergence, (iii) whether the gain is explained by a learned redistribution policy rather than by a fixed heuristic or only by the local loss, and (iv) whether the resulting improvement is obtained with practical computational and communication cost. During evaluation, baseline methods are kept in their original algorithmic form, including their defining optimization or coordination mechanisms and their tuned settings, unless explicitly stated otherwise. By contrast, FedPolicy uses simple mean (FedAvg-style) aggregation and introduces adaptivity only through client-specific post-aggregation redistribution. This design isolates redistribution as the sole adaptive component of the proposed method, ensuring that the observed gains are not explained by any additional aggregation-side advantage, but by the client-specific adaptive model redistribution mechanism itself. Comprehensive experimental details, including implementation settings, training configurations, and data partitioning procedures, are provided in Appendix C.

### 4.1 Performance under Heterogeneous Data

We first evaluate FedPolicy on two primary axes: best top-1 accuracy under heterogeneous partitions and convergence speed toward useful operating points. We then analyze whether the observed gains are consistent with the proposed redistribution mechanism.

Table 2: Top-1 accuracy (%) on FMNIST, CIFAR-10, CIFAR-100, and ISIC2019 under size-varying Dirichlet label skew. Lower $\beta$ indicates stronger heterogeneity. Each experiment was conducted using three independent random seeds (0, 1, and 2), and the reported values denote the mean $\pm$ standard deviation of the peak global test accuracy across these runs. The best result in each column is highlighted in **bold**.

| Method | FMNIST | | | CIFAR-10 | | | CIFAR-100 | | | ISIC2019 | | |
|---|---|---|---|---|---|---|---|---|---|---|---|---|
| | $\beta$=0.3 | $\beta$=0.1 | $\beta$=0.05 | $\beta$=0.3 | $\beta$=0.1 | $\beta$=0.05 | $\beta$=0.3 | $\beta$=0.1 | $\beta$=0.05 | $\beta$=0.3 | $\beta$=0.1 | $\beta$=0.05 |
| FedAvg | 82.51±0.38 | 78.75±0.52 | 73.11±0.89 | 64.76±0.61 | 31.25±1.14 | 16.11±1.52 | 49.83±0.67 | 43.92±0.95 | 37.22±1.38 | 55.82±0.86 | 49.24±1.09 | 48.68±1.21 |
| FedProx | 82.65±0.35 | 78.89±0.49 | 73.72±0.87 | 64.82±0.58 | 33.79±1.09 | 16.31±1.47 | 49.87±0.64 | 43.97±0.91 | 37.23±1.35 | 57.67±0.64 | 49.68±1.16 | 49.16±1.43 |
| FedNTD | 83.95±0.32 | 80.04±0.51 | 74.23±0.89 | 69.20±0.62 | 44.08±1.15 | 21.52±1.54 | 50.19±0.58 | 44.93±1.17 | 38.34±1.23 | 63.73±0.79 | 50.13±0.96 | 48.85±1.29 |
| Ditto | 83.13±0.41 | 78.15±0.59 | 73.23±0.96 | 65.00±0.67 | 31.25±1.21 | 17.18±1.61 | 50.00±0.68 | 37.64±1.08 | 28.96±1.52 | 61.83±0.82 | 52.36±1.09 | 50.31±1.27 |
| CCVR | 83.63±0.39 | 78.97±0.54 | 71.78±0.99 | 67.31±0.59 | 38.57±1.12 | 15.78±1.64 | 48.95±0.73 | 42.68±0.98 | 36.41±1.41 | 62.48±0.75 | 53.23±0.99 | 51.42±1.28 |
| FedSAM | 82.06±0.45 | 78.87±0.61 | 70.83±1.03 | 64.13±0.79 | 27.25±1.28 | 14.66±1.71 | 46.98±0.78 | 38.67±1.15 | 32.15±1.58 | 59.74±0.91 | 50.23±1.02 | 48.37±1.36 |
| FedFA | 83.31±0.43 | 77.48±0.63 | 74.59±0.92 | 64.51±0.72 | 28.31±1.25 | 17.35±1.55 | 49.27±0.75 | 37.78±1.11 | 31.74±1.53 | 65.33±0.59 | 56.90±1.13 | 52.73±1.17 |
| FedPart | 77.79±0.47 | 71.73±0.52 | 41.82±1.03 | 66.59±0.67 | 27.89±1.47 | 13.93±1.61 | 45.97±0.68 | 40.46±1.08 | 29.40±1.52 | 53.70±0.66 | 48.71±1.19 | 47.52±1.36 |
| FedCA | 84.64±0.56 | 81.24±0.47 | 78.89±0.84 | 73.46±0.52 | 59.01±1.48 | 51.62±1.76 | 50.65±0.78 | 45.62±0.98 | 39.41±1.29 | 66.92±0.31 | 58.18±0.84 | 54.51±0.91 |
| **FedPolicy** | **85.84±0.61** | **82.68±0.57** | **79.94±0.89** | **74.28±0.52** | **62.84±1.13** | **55.25±1.38** | **51.83±0.74** | **48.12±1.13** | **42.72±1.34** | **68.23±0.64** | **67.71±0.47** | **63.49±0.98** |

### 4.1.1 Accuracy under Label Heterogeneity

Table 2 compares FedPolicy against nine baselines on FMNIST, CIFAR-10, and CIFAR-100 and against the evaluated baselines on ISIC2019, using the same size-varying Dirichlet label-skew levels $\beta \in \{0.3, 0.1, 0.05\}$. These benchmarks span different class counts (8, 10, and 100), image modalities (grayscale, natural RGB images, and dermoscopic images), and task difficulties. Across all four datasets, FedPolicy achieves the best result in every reported setting. The advantage is modest when heterogeneity is mild and becomes more pronounced on the more challenging benchmarks and under stronger heterogeneity settings. On CIFAR-10, for example, the improvement over the strongest baseline increases from +1.12% at $\beta$=0.3 to +7.03% at $\beta$=0.05. A similar trend appears on CIFAR-100, where the margin increases from +2.33% to +8.40% as $\beta$ decreases. On FMNIST, the gains are smaller in relative terms but remain consistent across all settings, ranging from +1.33% to +1.77%.

The ISIC2019 results further test whether this behavior extends to a heterogeneous medical-image benchmark. FedPolicy reaches 68.23%, 67.71%, and 63.49% at $\beta$=0.3, 0.1, and 0.05, respectively. FedCA is the strongest non-FedPolicy baseline at all three heterogeneity levels, reaching 66.92%, 58.18%, and 54.51%. Relative to FedCA, FedPolicy improves accuracy by +1.31, +9.53, and +8.98 percentage points. FedFA is also competitive at $\beta$=0.3 and $\beta$=0.05, reaching $65.33 \pm 0.59\%$ and $52.73 \pm 1.17\%$, while FedPart reaches $53.70 \pm 0.66\%$, $48.71 \pm 1.19\%$, and $47.52 \pm 1.36\%$ across the three heterogeneity levels. Thus, the advantage again becomes more pronounced under the strongest heterogeneity setting. These controlled ISIC2019 runs are averaged over seeds $0, 1, 2$, and the small standard deviations of FedPolicy (0.64, 0.47, and 0.98) indicate stable behavior across the evaluated partitions.

Furthermore, Table 3 indicates that the gain of Fed-Policy is not solely explained by stronger local optimization. When relative improvement is computed against the best baseline at each heterogeneity level and then averaged across $\beta \in \{0.3, 0.1, 0.05\}$, Fed-Policy achieves 10.93% under CE and 4.73% under C3E. Concretely, the CE improvements are +3.07%, +10.42%, and +19.30%, while the C3E improvements are +0.68%, +6.49%, and +7.03%. FedPart is included under the same CE and C3E local-loss settings for this comparison. Together, these results show that stronger local optimization alone does not account for the gain and that adaptive sharing becomes particularly valuable under severe client drift.

Overall, Table 2 and Table 3 show that, even without a specialized aggregation rule, adaptive client-specific post-aggregation parameter sharing yields consistent gains over strong baselines across the

Table 3: Effect of local training loss on CIFAR-10 under size-varying Dirichlet label skew. Entries report peak Top-1 accuracy (%) for two client-side training objectives: CE (standard cross-entropy) and C3E (Confusion-Calibrated Cross-Entropy). Best result in each column is shown in **bold**.

| | $\beta$=0.3 | | $\beta$=0.1 | | $\beta$=0.05 | |
|---|---|---|---|---|---|---|
| Method | CE | C3E | CE | C3E | CE | C3E |
| FedAvg | 64.76 | 71.30 | 31.25 | 49.27 | 16.11 | 39.19 |
| FedProx | 64.82 | 72.10 | 33.79 | 44.20 | 16.31 | 40.12 |
| Ditto | 65.00 | 72.40 | 31.25 | 43.10 | 17.18 | 40.50 |
| CCVR | 67.31 | 73.78 | 38.57 | 51.80 | 15.78 | 48.30 |
| FedSAM | 54.13 | 61.80 | 27.25 | 38.70 | 14.66 | 46.10 |
| FedNTD | 69.20 | 72.98 | 44.08 | 53.77 | 21.52 | 32.54 |
| FedFA | 63.51 | 71.80 | 28.31 | 41.60 | 17.35 | 46.83 |
| FedPart | 66.59 | 72.94 | 23.08 | 53.21 | 13.93 | 22.86 |
| FedCA | 69.28 | 73.46 | 40.68 | 59.01 | 23.83 | 51.62 |
| **FedPolicy** | **71.41** | **74.28** | **48.67** | **62.84** | **28.43** | **55.25** |

evaluated size-varying Dirichlet label-skew settings, with larger benefits in regimes where uniform rebroadcast is least reliable. The local-training-loss comparison indicates a contribution beyond stronger client optimization alone, while the ISIC2019 results extend the evidence to a heterogeneous medical-image benchmark.

### 4.1.2 Convergence Behavior

Table 4 reports the first communication round at which each method reaches predefined accuracy thresholds on CIFAR-10 ($\beta$=0.3), while Figure 3 shows full learning curves under the harder $\beta$=0.1 setting. Together, these results evaluate convergence speed rather than only final accuracy.

FedPolicy reaches useful operating points earlier than all baselines. In Table 4, it reaches 40% at round 3 (vs. round 5 for FedCA, round 6 for CCVR, and round 19 for FedPart) and 60% at round 11 (vs. round 16 for FedCA, round 29 for FedNTD, and round 71 for FedPart). At the highest threshold, both FedPolicy and FedCA exceed 70%, but FedPolicy does so substantially earlier (round 31 vs. round 46), while FedPart does not reach this threshold within 105 rounds. These margins show that the gain is not limited to endpoint performance; it appears throughout training, especially in the early and mid stages.

Figure 3 confirms the same pattern under stronger heterogeneity ($\beta$=0.1): FedPolicy separates from FedCA within roughly 15–20 rounds and maintains a consistent gap thereafter. FedPart improves later in training but remains below FedPolicy across the reported trajectory. Thus, the improvement is trajectory-level, not a late-round fluctuation. Overall, these convergence results strengthen the main claim that client-specific post-aggregation sharing improves both final accuracy and training efficiency under heterogeneous data.

## 4.2 DRL
### Policy Behavior under Data Heterogeneity

Figures 4 and 5 show that the learned redistribution policy varies substantially across DRL controllers, with the differences becoming clearer under stronger heterogeneity. Under severe heterogeneity ($\beta = 0.05$), SAC and PPO are both strongly Backbone-dominant, allocating 83.0% and 93.0% of actions to Backbone sharing, respectively, whereas DQN remains close to a uniform allocation (34% Full, 35% Backbone, 32% Head). At $\beta = 0.1$, the separation is more pronounced:

Table 4: Convergence on CIFAR-10 ($\beta$=0.3): first round reaching each accuracy threshold (lower is better). "$\times$" indicates the threshold was not reached within 105 rounds.

| Method | $\geq 20\%$ | $\geq 40\%$ | $\geq 60\%$ | $\geq 70\%$ |
|---|---|---|---|---|
| FedAvg | 1 | 16 | 63 | $\times$ |
| FedProx | 1 | 14 | 57 | $\times$ |
| Ditto | 1 | 17 | 49 | $\times$ |
| CCVR | 2 | 6 | 33 | $\times$ |
| FedSAM | 1 | 27 | 61 | $\times$ |
| FedFA | 1 | 8 | 72 | $\times$ |
| FedNTD | 2 | 16 | 29 | $\times$ |
| FedPart | 1 | 19 | 71 | $\times$ |
| FedCA | 1 | 5 | 16 | 46 |
| **FedPolicy** | **1** | **3** | **11** | **31** |

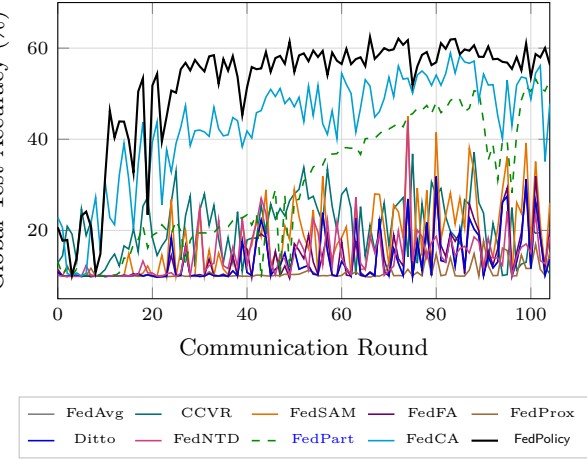

Figure 3: Global test accuracy over communication rounds on CIFAR-10 under moderate heterogeneity ($\beta$=0.1). FedPolicy consistently achieves higher accuracy and faster convergence compared to all baselines.

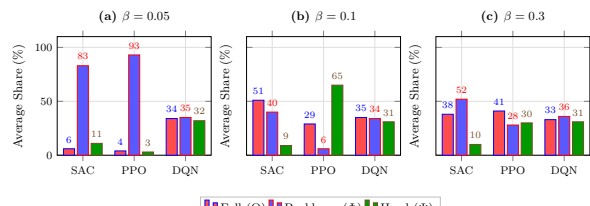

Figure 4: Action proportions for DRL controllers.

SAC follows a structured mixed policy (51% Full, 40% Backbone, 9% Head), PPO shifts to a Head-heavy pattern (65.0% Head), and DQN again remains near-uniform (35%/34%/31%). Under milder heterogeneity

($\beta = 0.3$), SAC still places the largest share on Backbone updates (52%), while PPO becomes more mixed and DQN continues to show little specialization.

The temporal trajectories in Figure 5 are consistent with the same pattern: SAC converges to a clearer redistribution preference over training, PPO varies more across regimes, and DQN remains comparatively diffuse. A broader controller comparison and additional policy analyses are provided in Appendix C.3.

Taken together, the evidence suggests that the benefit of FedPolicy depends not merely on using a DRL controller, but on whether that controller can learn a heterogeneity-aware and stable redistribution policy.

## 4.3 Ablation Studies

We next examine which components of FedPolicy are responsible for the observed gains. The ablations address four questions: which state signals are most informative for the controller, how the reward should be defined, whether the benefit comes from a learned policy rather than a fixed sharing rule, and whether the method remains effective under different local loss functions.

### 4.3.1 Learned versus Static Sharing Strategies

Table 5 compares five DRL controllers with four static parameter-sharing heuristics on CIFAR-10 under deterministic fixed-class-support splits and size-varying Dirichlet label skew. All entries in this controller-ablation study use C3E as the local training loss, with the same model architecture,

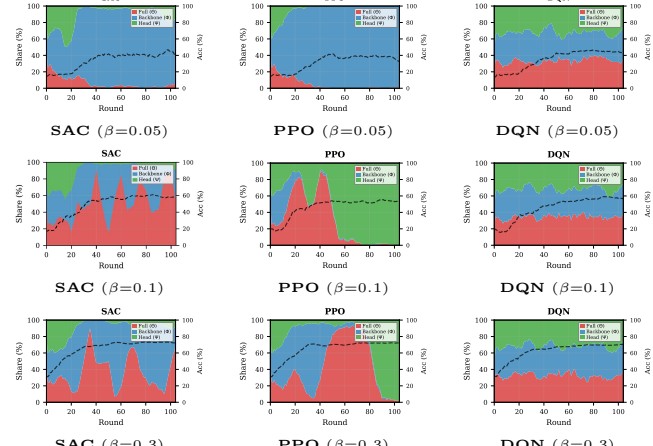

Figure 5: Policy evolution for SAC, PPO, and DQN across three Dirichlet levels. Each panel shows the action proportions over communication rounds together with the corresponding global test-accuracy trajectory (black dashed curve, right axis).

communication budget, client partitions, and evaluation protocol; thus, Full-Only corresponds to uniform full-model redistribution under this C3E setting, rather than the CE-based FedAvg baseline reported in Table 2. We also include a contextual-bandit controller that uses the same state and action space but learns only from immediate rewards, without bootstrapping future value. The goal is to test whether gains come from learning the policy itself, rather than from any fixed sharing rule.

This study directly examines whether a learned redistribution policy is needed beyond simple static or random choices. If one fixed rule were sufficient, Full-Only, Backbone-Only, Head-Only, or Random redistribution would remain competitive across heterogeneity regimes; instead, the comparison below shows that the preferred action changes with the client distribution and that learned controllers become increasingly useful as heterogeneity strengthens.

Under fixed-class partitions, learned controllers are consistently competitive and become clearly superior as class support narrows. At fc=6 and fc=4, TRPO reaches 78.78% and 75.41%, respectively, exceeding the best static heuristics (77.12% and 73.78%). At the hardest setting (fc=2), DQN attains 55.35%, outperforming the best static baseline (Random, 52.81%) by +2.54 points. Head-Only remains weak (30.48%), indicating that classifier-only

Table 5: Controller ablation on CIFAR-10 under two client-partition settings using C3E as the local training loss for all entries. **Left:** deterministic fixed-class-support splits (controlled by classes/client). **Right:** size-varying Dirichlet label-skew splits (controlled by the concentration parameter). Full-Only denotes uniform full-model redistribution under this matched C3E setting. Best result in each column is shown in **bold**.

| | *Fixed-Class (Deterministic)* | | | *Dirichlet (Probabilistic)* | | |
|---|---|---|---|---|---|---|
| Controller | fc=6 | fc=4 | fc=2 | $\beta$=0.3 | $\beta$=0.1 | $\beta$=0.05 |
| *Static Heuristics* | | | | | | |
| Full-Only | 77.12 | 73.64 | 49.87 | 72.67 | 54.94 | 42.42 |
| Backbone-Only | 76.32 | 73.78 | 48.22 | **74.75** | 56.40 | 41.17 |
| Head-Only | 69.64 | 59.85 | 30.48 | 59.16 | 25.22 | 22.57 |
| Random Policy | 73.40 | 72.88 | 52.81 | 72.37 | 58.99 | 42.07 |
| *Contextual Bandit Baseline* | | | | | | |
| Contextual Bandit | 56.17 | 53.51 | 42.44 | 50.84 | 44.06 | 39.32 |
| *DRL Algorithms* | | | | | | |
| SAC | 77.59 | 74.22 | 54.92 | 74.28 | 62.84 | **54.38** |
| TRPO | **78.78** | 75.41 | 50.39 | 73.88 | **64.32** | 54.34 |
| A2C | 78.60 | 72.96 | 54.97 | 73.80 | 63.26 | 52.85 |
| PPO | 76.33 | 72.16 | 51.00 | 73.81 | 58.52 | 52.56 |
| DQN | 77.21 | 73.21 | **55.35** | 72.58 | 62.19 | 50.66 |

transfer is insufficient under severe class scarcity.
The contextual-bandit controller remains below both the best static heuristic and the best DRL controller in all fixed-class settings, suggesting that immediate state-conditioned reward fitting is not sufficient to recover the stronger sequential controllers.

The same trend appears more strongly under size-varying Dirichlet label skew. At $\beta$=0.3, Backbone-Only reaches 74.75%, slightly above the best DRL result (SAC, 74.28%), indicating that in this milder regime the additional controller is not justified by accuracy alone. As heterogeneity increases, however, the benefit of learned policies becomes substantial: at $\beta$=0.1, TRPO reaches 64.32% versus 58.99% for the best static rule (+5.33), and at $\beta$=0.05, SAC reaches 54.38% versus 42.42% (+11.96). No single static heuristic is consistently best across settings. Notably, all five DRL methods outperform the best static rule at $\beta$=0.05. The contextual bandit reaches 50.84%, 44.06%, and 39.32% at $\beta$=0.3, 0.1, and 0.05, respectively, and therefore does not match either the strongest static heuristic or the sequential DRL controllers. This gap indicates that observing client state is useful but insufficient by itself; the advantage of the learned RL policy appears when redistribution decisions have delayed effects through later local updates and subsequent global aggregation. Thus, the added algorithmic complexity is most justified in moderate-to-severe heterogeneous regimes, where the preferred redistribution action changes across clients and rounds; under mild heterogeneity, a simple static heuristic may be sufficient. These results provide empirical evidence that state-dependent learned redistribution is useful in the stronger heterogeneous regimes, but they do not establish that the controller is optimal or that it minimizes redistribution mismatch at every round.

### 4.4 State Representation and Reward Design

Table 6 isolates the contributions of state features and reward design. For state representation, parameter divergence is the most consistently useful signal: configurations without it show the largest drops (e.g., 74.27% $\rightarrow$ 71.56% at $\beta$=0.3, and 62.59% $\rightarrow$ 60.44% at $\beta$=0.1). This supports the view that local–global mismatch is a primary driver of redistribution decisions.

No single state variant dominates all regimes. At $\beta$=0.3, NCF is best (74.27%); at $\beta$=0.1, the full state is best (62.59%); and at $\beta$=0.05, NCFNLA is best (52.54%). This pattern suggests that informative signals shift with heterogeneity severity, while divergence remains broadly valuable.

For reward design, the proposed combined reward is best at $\beta$=0.3 and $\beta$=0.1 (73.03%, 61.84%), and close to the best at $\beta$=0.05 (vs. 51.94% for Global-only). Overall, balancing client-level and global-progress signals yields the most reliable behavior across settings.

Table 6: State and reward ablation for FedPolicy on CIFAR-10 across heterogeneity levels ($\beta$). The full state (All) includes $\vec{C}_k$, $\mathrm{acc}_k$, ACC, and $\|\Delta\theta\|$. Best results per column are in **bold**.

| Configuration | $\beta$=0.3 | $\beta$=0.1 | $\beta$=0.05 |
|---|---|---|---|
| *State Representation* | | | |
| All: $[\vec{C}_k \oplus \mathrm{acc}_k \oplus \mathrm{ACC} \oplus \|\Delta\theta\|]$ | 73.83 | **62.59** | 51.80 |
| NCF: $[\mathrm{acc}_k \oplus \mathrm{ACC} \oplus \|\Delta\theta\|]$ | **74.27** | 62.20 | 50.95 |
| NCFNLA: $[\mathrm{ACC} \oplus \|\Delta\theta\|]$ | 73.08 | 62.45 | **52.54** |
| OCMLA: $[\vec{C}_k \oplus \mathrm{acc}_k \oplus \mathrm{ACC}]$ | 71.56 | 60.44 | 49.92 |
| OPD: $[\vec{C}_k \oplus \mathrm{acc}_k \oplus \|\Delta\theta\|]$ | 73.27 | 61.95 | 52.20 |
| *Reward Strategy* | | | |
| Local-only $(\lambda_{\mathrm{local}}\Delta\mathrm{acc}_k)$ | 72.67 | 61.21 | 51.17 |
| Global-only $(\lambda_{\mathrm{glob}}\Delta\mathrm{ACC})$ | 72.43 | 61.72 | **51.94** |
| Proposed $(\lambda_{\mathrm{local}}\Delta\mathrm{acc}_k + \lambda_{\mathrm{glob}}\Delta\mathrm{ACC})$ | **73.03** | **61.84** | 51.69 |

### 4.5 Practical Efficiency

A final question is whether the gain of FedPolicy is obtained at a meaningful systems cost. The answer is favorable.

#### 4.5.1 Cost to Accuracy Trade-off

Table 7 summarizes the cumulative communication and computation required to first reach a sequence of target accuracies on CIFAR-10 under $\beta$=0.1. This analysis measures practical efficiency, not only best accuracy. For most baselines, communication per round is fixed because each selected client exchanges the full model in both directions; FedPart instead follows its scheduled partial-layer exchange. For FedPolicy, communication is asymmetric: clients still upload full models, while only the downlink is selective. Consequently, cumulative communication gains arise from the combination of reduced downlink payload and faster convergence in rounds.

Table 7: Cost-to-accuracy analysis on CIFAR-10 ($\beta$=0.1). For each method, we report the first round reaching 20%, 30%, 40%, and 50% global test accuracy, together with cumulative communication (GB) and computation time (minutes) at that point. "$\times$" indicates a threshold not reached within 105 rounds. Lower is better. Best values are in **bold**.

| Method | Round ($\downarrow$) | | | | Comm. (GB) ($\downarrow$) | | | | Comp. (min) ($\downarrow$) | | | | Best (%) |
| | 20% | 30% | 40% | 50% | 20% | 30% | 40% | 50% | 20% | 30% | 40% | 50% | |
|---|---|---|---|---|---|---|---|---|---|---|---|---|---|
| FedAvg | 43 | 80 | $\times$ | $\times$ | 36.7 | 67.5 | $\times$ | $\times$ | 15.3 | 28.2 | $\times$ | $\times$ | 31.9 |
| FedProx | 40 | 74 | $\times$ | $\times$ | 33.3 | 61.7 | $\times$ | $\times$ | 14.2 | 26.1 | $\times$ | $\times$ | 31.9 |
| FedSAM | 24 | 56 | 74 | $\times$ | 20.8 | 47.5 | 62.5 | $\times$ | 8.1 | 18.5 | 24.3 | $\times$ | 45.1 |
| FedFA | 68 | 101 | $\times$ | $\times$ | 57.5 | 85.0 | $\times$ | $\times$ | 31.6 | 46.7 | $\times$ | $\times$ | 31.9 |
| CCVR | 2 | 25 | $\times$ | $\times$ | 2.5 | 21.7 | $\times$ | $\times$ | 1.6 | 13.5 | $\times$ | $\times$ | 37.2 |
| FedNTD | 30 | 74 | 74 | $\times$ | 25.8 | 62.5 | 62.5 | $\times$ | 10.8 | 26.1 | 26.1 | $\times$ | 44.1 |
| Ditto | 43 | 80 | $\times$ | $\times$ | 36.7 | 67.5 | $\times$ | $\times$ | 15.3 | 28.2 | $\times$ | $\times$ | 31.9 |
| FedCA | 1 | 13 | 21 | 87 | 0.8 | 10.8 | 17.5 | 72.5 | 0.3 | 4.6 | 7.4 | 30.5 | 59.0 |
| FedPart | 18 | 52 | 65 | 88 | 4.3 | 10.2 | 10.3 | **11.2** | 6.3 | 18.3 | 22.9 | 31.0 | 53.2 |
| **FedPolicy (Ours)** | **1** | **11** | **11** | **23** | **0.7** | **7.5** | **7.5** | 12.1 | **0.3** | **3.7** | **3.7** | **5.6** | **63.5** |

The results show that FedPolicy is markedly more efficient in reaching useful accuracy levels and is also faster than FedCA at the shared thresholds. It reaches 30% accuracy in 11 rounds using 7.5 GB and 3.7 minutes, whereas FedCA requires 13 rounds, 10.8 GB, and 4.6 minutes. The gap becomes larger at higher thresholds: FedPolicy reaches 40% accuracy in 11 rounds, compared with 21 rounds for FedCA, and reaches 50% by round 23 with 12.1 GB of communication and 5.6 minutes of computation. FedCA and FedPart also reach 50% within the budget, but only at rounds 87 and 88, respectively. FedPart's scheduled partial updates give low cumulative communication at the 50% threshold, but this comes after many more rounds and substantially higher computation time than FedPolicy. FedPolicy ultimately reaches 63.5% accuracy.

These results show that the benefit of FedPolicy is not limited to improved best accuracy. The method reaches useful operating points much earlier and at substantially lower cumulative cost. In particular, Head-only broadcasts reduce the downlink payload from 42.66 MB to 0.02 MB for a given client, which explains the large savings in the early training phase. Taken together, the computational and communication analyses indicate that adaptive post-aggregation redistribution is practically lightweight while yielding a substantially better cost-to-accuracy trade-off under heterogeneous data.

A detailed discussion, including key observations, privacy considerations, and limitations, is provided in Appendix D.

## 5 Conclusion and Future Directions

This paper examined a common assumption in federated learning: the aggregated global model is shared back to all clients in the same form at every round. Under statistical heterogeneity, such uniform reuse can induce negative transfer by disrupting locally adapted representations. To address this limitation, we proposed FedPolicy, a policy-guided selective redistribution mechanism that learns whether each client should receive the full model, the backbone, or the classifier head after aggregation. Across FMNIST, CIFAR-10, CIFAR-100, and ISIC2019, FedPolicy consistently outperformed strong baselines, with the advantage becoming more pronounced in the more challenging heterogeneous settings. Averaged over all four datasets and heterogeneity levels, FedPolicy achieves an average relative gain of approximately 5.85% over the strongest baseline. On CIFAR-10, for example, the relative improvement grew from +1.12% at $\beta$=0.3 to +7.03% at $\beta$=0.05, while on ISIC2019 it reached +16.47% under severe heterogeneity. Beyond improving final accuracy, the method also accelerated convergence, reaching 60% accuracy on CIFAR-10 at $\beta$=0.3 in 11 rounds compared with 29 rounds for FedNTD, and attained the same operating point under $\beta$=0.1 with substantially lower communication cost than FedAvg. These gains were obtained with negligible additional overhead, with the controller contributing only 0.021 s per round. Taken together, these results support the central claim of the paper: client-specific post-aggregation redistribution is an important and underexplored design dimension in federated learning under heterogeneity.

In future, we plan to extend the current discrete action space to finer-grained sharing strategies, such as layer-wise selection or continuous mixing. It would also be valuable to examine the method under broader deployment conditions, including asynchronous participation, system heterogeneity, and time-varying client distributions. It would also be interesting to study alternative adaptive control mechanisms, such as supervised action predictors or threshold-based decision rules, within the same redistribution framework.

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

## Appendix

## A    Related Work

Statistical heterogeneity remains a central challenge in federated learning (FL), motivating extensive research on improving convergence stability and generalization under non-IID data distributions. Early work such as FedAvg McMahan et al. (2017) establishes the standard paradigm of local training followed by server-side aggregation, but suffers from client drift when local objectives differ significantly. To mitigate this, methods such as FedProx Li et al. (2020) introduce proximal regularization to constrain local updates, while SCAFFOLD Karimireddy et al. (2020) employs control variates to correct drift induced by heterogeneous data. Related server-side optimization strategies further stabilize training through normalized or adaptive aggregation rules. FedACG Kim et al. (2024) follows this direction by accelerating client gradients: the server broadcasts a lookahead global model and local training is regularized toward this overshot model, improving communication efficiency and stability under low participation. Despite their effectiveness, these approaches implicitly assume that the aggregated global model can be uniformly and fully reapplied to all clients, leaving the post-aggregation client update step fixed and unmodeled.

Personalized federated learning (PFL) relaxes the single-global-model assumption by allowing client-specific parameters through multi-task learning, clustering, mixture models, or partial parameter sharing Arivazhagan et al. (2019); Collins et al. (2021). A common design separates models into shared and private components, often retaining a global backbone while personalizing classifier heads. These approaches demonstrate improved client-level performance by reducing negative transfer across heterogeneous distributions. However, personalization structures are typically predefined and static, and optimization is driven by client-local objectives. As a result, PFL methods do not explicitly model how client update decisions influence future aggregation quality or global convergence behavior.

Another line of work improves robustness under heterogeneity by adapting server-side coordination mechanisms, including dynamic client weighting, similarity-aware aggregation, and fairness-oriented reweighting Li et al. (2021). More recently, reinforcement learning (RL) has been employed to automate coordination decisions, such as client selection or aggregation weight tuning, using validation feedback as reward Wang et al. (2020a). While these RL-based methods introduce adaptivity at the aggregation stage, they still apply the resulting global model back to clients using a uniform overwrite or interpolation rule, thereby leaving client-side initialization dynamics unaddressed.

Selective and partial parameter update strategies further demonstrate that *which* parameters are synchronized plays a crucial role in federated optimization. Recent work such as FedPart Wang et al. (2024a) shows that full-model synchronization can induce layer mismatch under heterogeneity, and proposes restricting updates to subsets of layers using predefined schedules. Related approaches freeze or alternate updates between backbone and head to improve stability and efficiency. While these methods highlight the importance of parameter subspace selection, update schedules are typically heuristic and shared across clients, rather than learned as a function of evolving client learning states.

Class imbalance and label skew further exacerbate heterogeneity in FL, motivating imbalance-aware objectives and reweighting strategies Zhang et al. (2022a). However, global class reweighting is often misaligned with client-specific label distributions, and privacy constraints limit direct access to global class statistics. Moreover, existing imbalance-aware methods primarily focus on improving local optimization rather than leveraging class-level error structure as a coordination signal for adaptive system-level control. FedCA is a recent confusion-aware FL method in this family. It introduces C3E as a confusion-calibrated local objective that uses each client's recurring class-confusion patterns to emphasize difficult class pairs during local training. FedCA further couples this local objective with a class-prioritized server aggregation rule, so that aggregation gives additional attention to classes that are more difficult under the current heterogeneous training dynamics.

For positioning, we emphasize a clear novelty boundary. Prior heterogeneity-aware studies have addressed confusion-aware local objectives and class-prioritized aggregation. The present paper does not reuse aggregation-side prioritization; instead, it addresses a different stage that remained fixed in those formulations: post-aggregation redistribution. By keeping aggregation standard (mean/FedAvg-style) and introducing adaptivity only in client-specific redistribution, the proposed formulation isolates the contribution and avoids conflating redistribution effects with aggregation complexity.

In contrast to prior work that emphasizes local optimization, aggregation weighting, personalization structure, or heuristic partial updates, this work focuses on an underexplored but consequential dimension of federated learning: *how heterogeneous clients should initialize from aggregated parameters after standard aggregation*. We retain standard aggregation operators (e.g., FedAvg) and instead learn a policy that controls post-aggregation client initialization (full model, backbone-only, or head-only updates). By leveraging confusion-aware local training signals that compactly characterize class-level struggle patterns, our approach enables principled policy learning for client-specific update control and explicitly models the long-term impact of initialization decisions on future aggregation quality.

# B  Detailed Theoretical Proofs

Table 8: Notation used in the theoretical proofs.

| Symbol | Description |
| --- | --- |
| $\mathbf{w}^t$ | Aggregated global model at communication round $t$ |
| $\tilde{\mathbf{w}}_k^t$ | Client-specific initialized model for client $k$ at round $t$ |
| $\boldsymbol{\delta}_k^t$ | Initialization difference, $\tilde{\mathbf{w}}_k^t - \mathbf{w}^t$ |
| $\bar{\boldsymbol{\delta}}^t$ | Weighted mean initialization difference, $\sum_k p_k \boldsymbol{\delta}_k^t$ |
| $D_t^2$ | Weighted initialization mismatch, $\sum_k p_k \|\boldsymbol{\delta}_k^t\|^2$ |
| $\mathcal{L}_k(\cdot)$ | Local objective of client $k$ |
| $\nabla \mathcal{L}_k(\cdot)$ | Gradient of the client-$k$ objective |
| $L$ | Smoothness constant of each $\mathcal{L}_k$ |
| $\mu$ | Block-wise strong convexity constant |
| $\epsilon_{\mathrm{drift}}$ | Upper bound on the relevant block gradient magnitude |
| $\mathbf{w}_{\mathsf{Full}}^t$ | Full global overwrite candidate |
| $\mathbf{w}_{\mathsf{Back}}^t$ | Backbone-only overwrite candidate |
| $\mathbf{w}_{\mathsf{Head}}^t$ | Head-only overwrite candidate |
| $\boldsymbol{\phi}^t$ | Global backbone parameters at round $t$ |
| $\boldsymbol{\psi}^t$ | Global head parameters at round $t$ |
| $\boldsymbol{\phi}_k^{t-1}$ | Previous local backbone parameters of client $k$ |
| $\boldsymbol{\psi}_k^{t-1}$ | Previous local head parameters of client $k$ |
| $\mathcal{U}$ | Neighborhood on which block-wise strong convexity holds |
| $\langle \cdot, \cdot \rangle$ | Standard Euclidean inner product |
| $\| \cdot \|$ | Standard Euclidean norm |

This appendix provides detailed proofs for the theoretical analysis presented in Section 3. The analysis isolates the effect of selective post-aggregation parameter sharing while keeping server aggregation unchanged and abstracting away the policy-learning dynamics of the RL controller.

## B.1 Proof Notation

Table 8 summarizes the symbols used in the proofs below.

## B.2 Assumptions Used in the Proofs

For completeness, we state below the assumptions used in the results of this appendix.

**Assumption 1** (Smoothness). *For each client $k$, the local objective $\mathcal{L}_k$ is $L$-smooth. Equivalently, for all $u, v \in \mathbb{R}^d$,*

$$\mathcal{L}_k(u) \leq \mathcal{L}_k(v) + \langle \nabla \mathcal{L}_k(v),\, u - v \rangle + \frac{L}{2}\|u - v\|^2.$$

**Assumption 2** (Block-wise strong convexity). *There exists a neighborhood $\mathcal{U}$ containing the candidate initialization points such that, for each client $k$:*

- *with the backbone fixed, the function*

$$h(\boldsymbol{\psi}) := \mathcal{L}_k([\boldsymbol{\phi}^t; \boldsymbol{\psi}])$$

  *is $\mu$-strongly convex in $\boldsymbol{\psi}$ on $\mathcal{U}$;*

- *with the head fixed, the function*

$$g(\boldsymbol{\phi}) := \mathcal{L}_k([\boldsymbol{\phi}; \boldsymbol{\psi}^t])$$

  *is $\mu$-strongly convex in $\boldsymbol{\phi}$ on $\mathcal{U}$.*

*Equivalently, for any differentiable $\mu$-strongly convex block-restricted function $q$ and any $u, v \in \mathcal{U}$,*

$$q(u) - q(v) \geq \langle \nabla q(v),\, u - v \rangle + \frac{\mu}{2}\|u - v\|^2.$$

**Assumption 3** (Block-gradient drift bound). *The candidate partially updated models satisfy the following block-gradient bounds:*

$$\|\nabla_{\boldsymbol{\psi}} \mathcal{L}_k(\mathbf{w}^t_{Back})\| \leq \epsilon_{\text{drift}}, \qquad \|\nabla_{\boldsymbol{\phi}} \mathcal{L}_k(\mathbf{w}^t_{Head})\| \leq \epsilon_{\text{drift}}.$$

The local comparison uses Assumptions 2 and 3. The stationarity characterization uses smoothness together with the explicitly stated one-step, full-participation update; it does not imply convergence of the learned policy.

## B.3 Block-wise Redistribution Comparison Bound

**Theorem 3** (Block-wise redistribution comparison bound). *Under Assumptions 2 and 3,*

$$\mathcal{L}_k(\mathbf{w}^t_{Full}) - \mathcal{L}_k(\mathbf{w}^t_{Back}) \geq \frac{\mu}{4}\|\boldsymbol{\psi}^t - \boldsymbol{\psi}_k^{t-1}\|^2 - \frac{\epsilon_{\text{drift}}^2}{\mu}, \tag{15}$$

$$\mathcal{L}_k(\mathbf{w}^t_{Full}) - \mathcal{L}_k(\mathbf{w}^t_{Head}) \geq \frac{\mu}{4}\|\boldsymbol{\phi}^t - \boldsymbol{\phi}_k^{t-1}\|^2 - \frac{\epsilon_{\text{drift}}^2}{\mu}. \tag{16}$$

*Proof.* We define the three candidate initialized models:

$$\mathbf{w}^t_{Full} = [\boldsymbol{\phi}^t; \boldsymbol{\psi}^t], \qquad \mathbf{w}^t_{Back} = [\boldsymbol{\phi}^t; \boldsymbol{\psi}_k^{t-1}], \qquad \mathbf{w}^t_{Head} = [\boldsymbol{\phi}_k^{t-1}; \boldsymbol{\psi}^t].$$

The proof compares Full separately with Back and Head, yielding action-specific comparison bounds.

**Step 1: Compare Full with Back.** Fix the backbone at $\phi^t$ and define the block-restricted function

$$h(\psi) := \mathcal{L}_k([\phi^t; \psi]).$$

By Assumption 2, the function $h$ is $\mu$-strongly convex in $\psi$ on $\mathcal{U}$. Therefore, for any $u, v \in \mathcal{U}$,

$$h(u) - h(v) \geq \langle \nabla h(v), \, u - v \rangle + \frac{\mu}{2}\|u - v\|^2. \tag{17}$$

Choose

$$u = \psi^t, \qquad v = \psi_k^{t-1}.$$

Then Eq. (17) gives

$$h(\psi^t) - h(\psi_k^{t-1}) \geq \langle \nabla h(\psi_k^{t-1}), \, \psi^t - \psi_k^{t-1} \rangle + \frac{\mu}{2}\|\psi^t - \psi_k^{t-1}\|^2.$$

Using the definition of $h$,

$$h(\psi^t) = \mathcal{L}_k(\mathbf{w}_{\mathsf{Full}}^t), \qquad h(\psi_k^{t-1}) = \mathcal{L}_k(\mathbf{w}_{\mathsf{Back}}^t),$$

and

$$\nabla h(\psi_k^{t-1}) = \nabla_\psi \mathcal{L}_k(\mathbf{w}_{\mathsf{Back}}^t).$$

Hence

$$\mathcal{L}_k(\mathbf{w}_{\mathsf{Full}}^t) - \mathcal{L}_k(\mathbf{w}_{\mathsf{Back}}^t) \geq \langle \nabla_\psi \mathcal{L}_k(\mathbf{w}_{\mathsf{Back}}^t), \, \psi^t - \psi_k^{t-1} \rangle + \frac{\mu}{2}\|\psi^t - \psi_k^{t-1}\|^2. \tag{18}$$

By Assumption 3,

$$\left\|\nabla_\psi \mathcal{L}_k(\mathbf{w}_{\mathsf{Back}}^t)\right\| \leq \epsilon_{\mathrm{drift}}.$$

Using the Cauchy–Schwarz inequality,

$$\langle \nabla_\psi \mathcal{L}_k(\mathbf{w}_{\mathsf{Back}}^t), \, \psi^t - \psi_k^{t-1} \rangle \geq -\left\|\nabla_\psi \mathcal{L}_k(\mathbf{w}_{\mathsf{Back}}^t)\right\| \cdot \left\|\psi^t - \psi_k^{t-1}\right\| \geq -\epsilon_{\mathrm{drift}}\left\|\psi^t - \psi_k^{t-1}\right\|.$$

Next we apply Young's inequality in the form

$$ab \leq \frac{\mu}{4}b^2 + \frac{1}{\mu}a^2.$$

Setting

$$a = \epsilon_{\mathrm{drift}}, \qquad b = \|\psi^t - \psi_k^{t-1}\|,$$

we obtain

$$-\epsilon_{\mathrm{drift}}\|\psi^t - \psi_k^{t-1}\| \geq -\frac{\mu}{4}\|\psi^t - \psi_k^{t-1}\|^2 - \frac{\epsilon_{\mathrm{drift}}^2}{\mu}.$$

Substituting this into Eq. (18), we get

$$\mathcal{L}_k(\mathbf{w}_{\mathsf{Full}}^t) - \mathcal{L}_k(\mathbf{w}_{\mathsf{Back}}^t) \geq \left(\frac{\mu}{2} - \frac{\mu}{4}\right)\|\psi^t - \psi_k^{t-1}\|^2 - \frac{\epsilon_{\mathrm{drift}}^2}{\mu}.$$

Therefore,

$$\mathcal{L}_k(\mathbf{w}_{\mathsf{Full}}^t) - \mathcal{L}_k(\mathbf{w}_{\mathsf{Back}}^t) \geq \frac{\mu}{4}\|\psi^t - \psi_k^{t-1}\|^2 - \frac{\epsilon_{\mathrm{drift}}^2}{\mu}. \tag{19}$$

**Step 2: Compare Full with Head.** Fix the head at $\boldsymbol{\psi}^t$ and define

$$g(\boldsymbol{\phi}) := \mathcal{L}_k([\boldsymbol{\phi}; \boldsymbol{\psi}^t]).$$

By Assumption 2, the function $g$ is $\mu$-strongly convex in $\boldsymbol{\phi}$ on $\mathcal{U}$. Thus, for any $u, v \in \mathcal{U}$,

$$g(u) - g(v) \geq \langle \nabla g(v), \, u - v \rangle + \frac{\mu}{2} \|u - v\|^2.$$

Choose

$$u = \boldsymbol{\phi}^t, \qquad v = \boldsymbol{\phi}_k^{t-1}.$$

Then

$$g(\boldsymbol{\phi}^t) - g(\boldsymbol{\phi}_k^{t-1}) \geq \langle \nabla g(\boldsymbol{\phi}_k^{t-1}), \, \boldsymbol{\phi}^t - \boldsymbol{\phi}_k^{t-1} \rangle + \frac{\mu}{2} \|\boldsymbol{\phi}^t - \boldsymbol{\phi}_k^{t-1}\|^2.$$

Using the definition of $g$,

$$g(\boldsymbol{\phi}^t) = \mathcal{L}_k(\mathbf{w}_{\mathsf{Full}}^t), \qquad g(\boldsymbol{\phi}_k^{t-1}) = \mathcal{L}_k(\mathbf{w}_{\mathsf{Head}}^t),$$

and

$$\nabla g(\boldsymbol{\phi}_k^{t-1}) = \nabla_{\boldsymbol{\phi}} \mathcal{L}_k(\mathbf{w}_{\mathsf{Head}}^t).$$

Hence

$$\mathcal{L}_k(\mathbf{w}_{\mathsf{Full}}^t) - \mathcal{L}_k(\mathbf{w}_{\mathsf{Head}}^t) \geq \langle \nabla_{\boldsymbol{\phi}} \mathcal{L}_k(\mathbf{w}_{\mathsf{Head}}^t), \, \boldsymbol{\phi}^t - \boldsymbol{\phi}_k^{t-1} \rangle + \frac{\mu}{2} \|\boldsymbol{\phi}^t - \boldsymbol{\phi}_k^{t-1}\|^2. \tag{20}$$

Again, by Assumption 3,

$$\left\| \nabla_{\boldsymbol{\phi}} \mathcal{L}_k(\mathbf{w}_{\mathsf{Head}}^t) \right\| \leq \epsilon_{\mathrm{drift}}.$$

Applying Cauchy–Schwarz and Young's inequality as above gives

$$\langle \nabla_{\boldsymbol{\phi}} \mathcal{L}_k(\mathbf{w}_{\mathsf{Head}}^t), \, \boldsymbol{\phi}^t - \boldsymbol{\phi}_k^{t-1} \rangle \geq -\frac{\mu}{4} \|\boldsymbol{\phi}^t - \boldsymbol{\phi}_k^{t-1}\|^2 - \frac{\epsilon_{\mathrm{drift}}^2}{\mu}.$$

Substituting this into Eq. (20), we obtain

$$\mathcal{L}_k(\mathbf{w}_{\mathsf{Full}}^t) - \mathcal{L}_k(\mathbf{w}_{\mathsf{Head}}^t) \geq \frac{\mu}{4} \|\boldsymbol{\phi}^t - \boldsymbol{\phi}_k^{t-1}\|^2 - \frac{\epsilon_{\mathrm{drift}}^2}{\mu}. \tag{21}$$

Equations (19) and (21) establish the desired block-wise comparison bounds. If $\|\boldsymbol{\psi}^t - \boldsymbol{\psi}_k^{t-1}\| > 2\epsilon_{\mathrm{drift}}/\mu$, then the right-hand side of the lower bound for $\mathbf{w}_{\mathsf{Back}}^t$ is strictly positive, implying $\mathcal{L}_k(\mathbf{w}_{\mathsf{Back}}^t) < \mathcal{L}_k(\mathbf{w}_{\mathsf{Full}}^t)$. Similarly, if $\|\boldsymbol{\phi}^t - \boldsymbol{\phi}_k^{t-1}\| > 2\epsilon_{\mathrm{drift}}/\mu$, then the right-hand side of the lower bound for $\mathbf{w}_{\mathsf{Head}}^t$ is strictly positive, implying $\mathcal{L}_k(\mathbf{w}_{\mathsf{Head}}^t) < \mathcal{L}_k(\mathbf{w}_{\mathsf{Full}}^t)$. This completes the proof.

### B.4 Stationarity Characterization under Selective-Initialization Error

We next prove Theorem 2. The redistribution error is defined from the actual aggregated update, and the one-round descent inequality is derived explicitly.

*Proof of Theorem 2.* Because each $\mathcal{L}_k$ is $L$-smooth and the weights form a convex combination, $f = \sum_k p_k \mathcal{L}_k$ is also $L$-smooth. Define the gradient perturbation

$$\mathbf{e}_t := \sum_{k=1}^{N} p_k \left( \nabla \mathcal{L}_k(\tilde{\mathbf{w}}_k^t) - \nabla \mathcal{L}_k(\mathbf{w}^t) \right) - \frac{\bar{\boldsymbol{\delta}}^t}{\eta}. \tag{22}$$

Using $\tilde{\mathbf{w}}_k^t = \mathbf{w}^t + \boldsymbol{\delta}_k^t$ and $\sum_k p_k = 1$, Eq. (13) becomes exactly

$$\mathbf{w}^{t+1} = \mathbf{w}^t + \bar{\boldsymbol{\delta}}^t - \eta \sum_k p_k \nabla \mathcal{L}_k(\tilde{\mathbf{w}}_k^t) \tag{23}$$

$$= \mathbf{w}^t - \eta \left( \nabla f(\mathbf{w}^t) + \mathbf{e}_t \right). \tag{24}$$

Thus $\mathbf{e}_t$ is not an assumed drift variable; it is the exact perturbation induced by evaluating client gradients at selectively initialized models and by averaging the retained parameter offsets.

Let $\mathbf{g}_t := \nabla f(\mathbf{w}^t)$. Smoothness of $f$, Eq. (24), and $L\eta \leq 1$ give

$$f(\mathbf{w}^{t+1}) \leq f(\mathbf{w}^t) - \eta\langle\mathbf{g}_t, \mathbf{g}_t + \mathbf{e}_t\rangle + \frac{L\eta^2}{2}\|\mathbf{g}_t + \mathbf{e}_t\|^2 \tag{25}$$

$$\leq f(\mathbf{w}^t) - \eta\langle\mathbf{g}_t, \mathbf{g}_t + \mathbf{e}_t\rangle + \frac{\eta}{2}\|\mathbf{g}_t + \mathbf{e}_t\|^2 \tag{26}$$

$$= f(\mathbf{w}^t) - \frac{\eta}{2}\|\mathbf{g}_t\|^2 + \frac{\eta}{2}\|\mathbf{e}_t\|^2. \tag{27}$$

It remains to bound the perturbation. By $\|a + b\|^2 \leq 2\|a\|^2 + 2\|b\|^2$, Jensen's inequality, and gradient Lipschitzness,

$$\|\mathbf{e}_t\|^2 \leq 2\left\|\sum_k p_k\left(\nabla\mathcal{L}_k(\tilde{\mathbf{w}}_k^t) - \nabla\mathcal{L}_k(\mathbf{w}^t)\right)\right\|^2 + \frac{2}{\eta^2}\|\bar{\boldsymbol{\delta}}^t\|^2 \tag{28}$$

$$\leq 2\sum_k p_k\left\|\nabla\mathcal{L}_k(\tilde{\mathbf{w}}_k^t) - \nabla\mathcal{L}_k(\mathbf{w}^t)\right\|^2 + \frac{2}{\eta^2}\|\bar{\boldsymbol{\delta}}^t\|^2 \tag{29}$$

$$\leq 2L^2 D_t^2 + \frac{2}{\eta^2}\|\bar{\boldsymbol{\delta}}^t\|^2. \tag{30}$$

Rearranging Eq. (27), summing from $t = 0$ to $T - 1$, using $f(\mathbf{w}^T) \geq f^\star$, and dividing by $\eta T/2$ yield

$$\frac{1}{T}\sum_{t=0}^{T-1}\|\nabla f(\mathbf{w}^t)\|^2 \leq \frac{2(f(\mathbf{w}^0) - f^\star)}{\eta T} + \frac{1}{T}\sum_{t=0}^{T-1}\|\mathbf{e}_t\|^2. \tag{31}$$

Substitution of Eq. (30) gives Eq. (14) and completes the proof.

## C  Additional Experimental Evaluation

This appendix collects supporting experiments that complement the main paper: implementation details, controller behavior analysis, local-training-loss comparison, unified controller-sensitivity analysis, client-dropout robustness, aggregation-compatibility evaluation, and the overhead analysis.

### C.1  Experimental Setup

#### C.1.1  Implementation Details

All methods are implemented in `PyTorch` using Python 3.9. We evaluate on four image classification benchmarks of increasing difficulty: Fashion-MNIST (FMNIST) Xiao et al. (2017), which contains 10 classes of $28 \times 28$ grayscale images with 70K total samples; CIFAR-10 Krizhevsky (2009), which comprises 10 classes of $32 \times 32$ RGB images with 60K samples; CIFAR-100 Krizhevsky (2009), which consists of 100 classes of $32 \times 32$ RGB images with 60K samples; and ISIC2019, an eight-class dermoscopic skin-lesion classification benchmark included through FLamby (du Terrail et al., 2022). For FMNIST, we use LeNet-5 LeCun et al. (1998) as the base model. For CIFAR-10, CIFAR-100, and ISIC2019, we use ResNet-18 He et al. (2016) as the feature extractor, followed by a linear classifier head $\Psi$ with $C \times 512 + C$ parameters, where $C$ denotes the number of classes. The ResNet-18 backbone contains $|\Phi| = 11,176,512$ parameters. For CIFAR-10, the classifier head contains $|\Psi| = 5,130$ parameters, accounting for only $\approx 0.046\%$ of the full model. In float32 precision, the complete ResNet-18-based model occupies $42.66\,\text{MB}$, whereas the classifier head requires only $0.02\,\text{MB}$. This pronounced backbone-head asymmetry is a key architectural property underlying our design. Because the classifier head constitutes only a negligible fraction of the total parameter budget, it enables selective global parameter sharing with fine-grained control over the balance between transferable global representations and client-specific classifier adaptation.

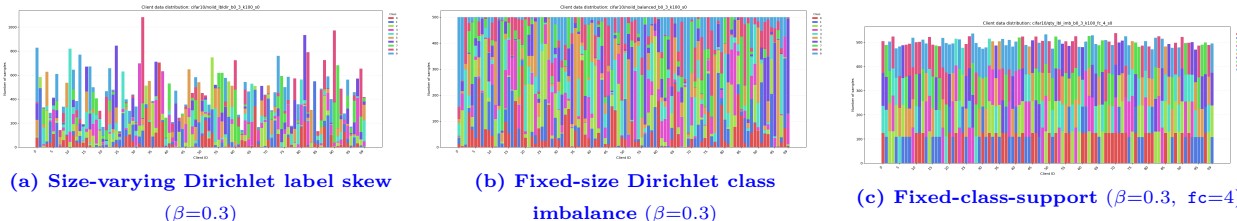

**(a) Size-varying Dirichlet label skew** ($\beta$=0.3)

**(b) Fixed-size Dirichlet class imbalance** ($\beta$=0.3)

**(c) Fixed-class-support** ($\beta$=0.3, `fc`=4)

Figure 6: CIFAR-10 client data distributions under three partition conventions. In the size-varying Dirichlet label-skew partition used in most of our experiments, the allocation changes both class composition and the number of training samples assigned to each client. In the fixed-size Dirichlet class-imbalance convention, each client has exactly 500 training samples and only the class composition varies. The fixed-class-support partition with $\beta$=0.3 and `fc`=4 assigns each client data from at most four classes, creating explicit class absence. Thus, identical Dirichlet concentration values can correspond to materially different federated data distributions when the client-size constraint differs, and fixed-class-support splits represent a separate heterogeneity protocol.

### C.1.2 Data Partitioning

We use two primary heterogeneous data-generation protocols in the experiments: size-varying Dirichlet label skew and fixed-class-support heterogeneity. The size-varying Dirichlet protocol is used for most main and appendix experiments, while the fixed-class-support protocol provides an additional stress test with explicit class absence.

- **Size-varying Dirichlet label skew.** We simulate label-distribution skew across clients using a Dirichlet-based partitioning scheme Hsu et al. (2019) with concentration parameter $\beta \in \{0.3, 0.1, 0.05\}$. Smaller $\beta$ values induce more severe heterogeneity by producing increasingly imbalanced class proportions across clients, whereas larger $\beta$ values yield more balanced local distributions. To ensure fair comparison, all methods use the same client partitions generated with a fixed random seed.

- **Fixed-class-support heterogeneity.** To evaluate robustness under explicit class absence, we construct partitions in which each client is assigned data from at most $\texttt{fc} \in \{6, 4, 2\}$ classes. Smaller `fc` values therefore correspond to more extreme heterogeneity, as clients become increasingly specialized to a limited subset of classes.

- **Fixed-size Dirichlet class imbalance.** For contextual comparison with the external CIFAR-10 result referenced in the reviewer comment, we also visualize a fixed-size Dirichlet partition convention. This partition fixes the number of training samples per client and varies only class composition; it is not used as the main evaluation protocol in our experiments.

The two primary study protocols provide complementary views of statistical heterogeneity: the size-varying Dirichlet scheme models stochastic variation in both label proportions and realized client sample counts, whereas the fixed-class-support setting imposes a more structured and severe form of class-distribution skew.

Figure 6 highlights an important reproducibility detail. Our reported Dirichlet setting does not enforce equal numbers of samples per client; as a result, $\beta$ controls label skew while the realized partition also contains quantity skew. This setting is therefore a stronger heterogeneity stress test than a fixed-size Dirichlet partition that uses the same $\beta$ value while keeping every client at the same sample count. The fixed-class-support example illustrates the second primary study protocol, where heterogeneity is induced by restricting the class support available to each client. We therefore compare baselines only under matched partitions generated by our data pipeline.

Table 9: Effect of partition convention on CIFAR-10. Entries report peak global Top-1 accuracy (%) under three client-partition conventions. All runs use the same evaluation budget and sampling setup (100 clients, 10 clients per round, 105 rounds, five local epochs, batch size 64, and seed 0), but this comparison uses its own training profile rather than the profile used in the main comparison table: ResNet-18, SGD with learning rate 0.01, momentum 0.9, learning-rate decay 0.998, and weight decay $5\times10^{-4}$. Method-specific objectives, regularizers, and controllers are kept according to each method's configured setting. Best result in each column is shown in **bold**.

| Method | Size-varying Dirichlet $\beta$=0.3 | Fixed-size Dirichlet $\beta$=0.3 | Fixed-class support fc=4 |
|---|---|---|---|
| FedAvg | 65.07 | 72.71 | 72.16 |
| FedProx | 56.93 | 74.17 | 72.08 |
| FedNTD | 70.38 | 74.72 | 75.11 |
| Ditto | 55.47 | 74.18 | 72.27 |
| CCVR | 64.10 | 73.72 | 72.87 |
| FedSAM | 69.58 | 74.68 | 75.22 |
| FedFA | 65.16 | 71.74 | 71.72 |
| FedPart | 66.81 | 71.19 | 58.67 |
| FedACG | 72.92 | 75.52 | 71.00 |
| FedCA | 74.39 | 75.10 | 75.23 |
| FedPolicy | **75.66** | **79.37** | **79.16** |

### C.1.3  Training Configuration

Unless otherwise stated, the main benchmark experiments use the following common training protocol. We simulate a federation of $N$=100 clients and use size-varying Dirichlet label skew with $\beta \in \{0.3, 0.1, 0.05\}$; lower $\beta$ indicates stronger label heterogeneity, and the size-varying protocol also induces client quantity skew. In each communication round, $C_r$=10 clients are sampled uniformly at random. Each selected client performs $E$=5 local epochs using SGD with learning rate $\eta$=$10^{-2}$, momentum 0.9, weight decay $5 \times 10^{-4}$, and batch size 64. The local learning rate is fixed at 0.01 after tuning over $[10^{-4}, 10^{-1}]$. Unless a table or subsection explicitly states a different evaluation budget, all methods are trained for $T$=105 communication rounds using the dataset-specific model described in Section C.1.1. For the baseline methods, we retain the local loss specified in their original formulations. When a local-training-loss comparison is reported, the table explicitly states whether CE or C3E is used for each method. For FedPolicy, we evaluate several local objectives, including standard cross-entropy (CE), Focal Loss (FL) Lin et al. (2017), logits-calibrated loss (FedLC) Zhang et al. (2022b), Class-Balanced loss (CB) Cui et al. (2019), Label Smoothing (LS) Szegedy et al. (2016), and Confusion-Calibrated Cross-Entropy (C3E) Chowdhury & Halder (2026). C3E is a client-side training loss that gives extra attention to class pairs that the local model repeatedly confuses. During local training, each client keeps a running estimate of its recent prediction confusions and uses this estimate to make commonly confused wrong classes more costly than rarely confused ones. In contrast, CE treats all wrong classes uniformly, while C3E uses the observed confusion pattern to change the local objective itself. The main comparison in Table 2 is evaluated over three independent random seeds $\{0, 1, 2\}$ and reported as mean $\pm$ standard deviation of the peak Top-1 global test accuracy. The supporting ablation, sensitivity, robustness, partition-convention, and compatibility studies are controlled single-seed runs using seed 0 unless their caption states otherwise; these tables and figures are therefore reported as peak Top-1 accuracy values rather than uncertainty estimates. A server-held-out validation split is used only to compute ACC($\cdot$) for controller state/reward; test labels are never used for controller training, model selection, or policy updates.

For server-side aggregation, each baseline follows the aggregation mechanism defined in its original paper, while FedPolicy uses a fixed mean/FedAvg-style rule. The aggregation-compatibility evaluation is the exception: there, the aggregation operator is varied deliberately and stated in Table 13. The default FedPolicy controller uses SAC Haarnoja et al. (2018) with the mixed local–global reward and the full state representation. The controller is parameterized by an MLP with hidden width 128, trained with learning rate $10^{-3}$, updated every three communication rounds, and optimized for five controller epochs per update. Alternative RL controllers (PPO, A2C, TRPO, and DQN) are evaluated only in the policy-ablation study. For ablation and supporting studies, these default optimizer, local-training, partitioning, and evaluation settings remain

unchanged unless the corresponding table caption or discussion explicitly states otherwise. All methods share identical random seeds, data partitions, and model initializations to ensure strict comparability.

### C.1.4 Baselines

We benchmark FedPolicy against eight popular FL baselines spanning diverse algorithmic families: FedAvg McMahan et al. (2017) (standard aggregation), FedProx Li et al. (2020) (drift correction), FedNTD Lee et al. (2022) (knowledge distillation), CCVR Luo et al. (2021) and FedFA Yang et al. (2023) (representation alignment), FedSAM Zhang et al. (2023) (sharpness-aware optimization), Ditto Li et al. (2021) (personalization), and FedCA Chowdhury & Halder (2026) (confusion-aware local optimization, originally proposed with class-prioritized aggregation). These baselines provide strong points of comparison because they address heterogeneity through different design principles—local regularization, representation alignment, personalization, modified optimization, or confusion-aware coordination—whereas FedPolicy targets the post-aggregation global knowledge transfer stage. All methods are evaluated under the same experimental protocol, with matched communication rounds and identical client partitions.

### C.2 Effect of Partition Convention on Empirical Difficulty

Table 9 shows that the same nominal $\beta$=0.3 value can lead to substantially different empirical difficulty depending on the partition convention. This comparison uses the hyperparameter profile stated in the caption and is intended to clarify the effect of the partition rule rather than replace the main benchmark protocol. For most methods, the size-varying Dirichlet setting is harder than the fixed-size Dirichlet setting because it combines label skew with quantity skew; for example, FedAvg changes from 72.71% under fixed-size Dirichlet to 65.07% under size-varying Dirichlet, and FedProx changes from 74.17% to 56.93%. The fixed-class-support protocol captures a different form of heterogeneity by imposing explicit class absence, which affects methods differently; for instance, FedPart drops to 58.67% under fc=4, whereas several other methods remain near the low-to-mid 70% range. The strongest non-FedPolicy baseline also depends on the partition convention: FedCA is strongest under size-varying Dirichlet and fixed-class-support heterogeneity (74.39% and 75.23%), while FedACG is strongest under fixed-size Dirichlet class imbalance (75.52%). FedPolicy is strongest under all three partition conventions, reaching 75.66%, 79.37%, and 79.16%, respectively. Thus, Table 9 complements Figure 6: $\beta$ alone is not sufficient to identify a matched FL benchmark unless the client-size constraint and partition rule are also specified.

### C.3 DRL-Policy Behavior over Data Heterogeneity

Figure 7 shows how the SAC controller changes its action preference over training. In the initial rounds (0–15), Head updates are most frequent (about 55%), indicating that the policy first favors local classifier adaptation while limiting global interference. As training proceeds, the share of Full updates rises to roughly 40–60%, whereas Backbone updates remain consistently active at around 30–40%. This suggests that the controller first allows client-specific specialization and later increases global coordination as training stabilizes.

Figures 8 and 9 show that the learned policy depends strongly on both the RL algorithm and the heterogeneity level. A clear distinction appears between the actor–critic methods (SAC, TRPO, PPO, and A2C) and DQN. While the actor–critic methods learn non-uniform and clearly structured policies, DQN remains close to a uniform allocation across Full, Backbone, and Head updates for all $\beta$ values. For example, its action split is 33.7%/34.6%/31.7% at $\beta$=0.05 and 32.9%/35.9%/31.2% at $\beta$=0.3, with entropy remaining near 1.0 throughout training. This indicates that DQN fails to specialize, even though such a uniform strategy remains reasonably competitive under mild heterogeneity.

Among the actor–critic methods, backbone-dominant policies emerge consistently as heterogeneity increases. Under severe heterogeneity ($\beta$=0.05), all four methods allocate most actions to Backbone updates, with shares ranging from 77.9% to 93.3%. TRPO is the most stable in this respect, maintaining a backbone share of 77.9%–80.4% across all three $\beta$ values and achieving the best or near-best accuracy throughout. SAC shows a similar tendency, favoring Backbone updates at $\beta$=0.05 (83.0%) while shifting toward more Full updates as heterogeneity weakens, and attains the highest accuracy at both $\beta$=0.05 (54.4%) and $\beta$=0.3

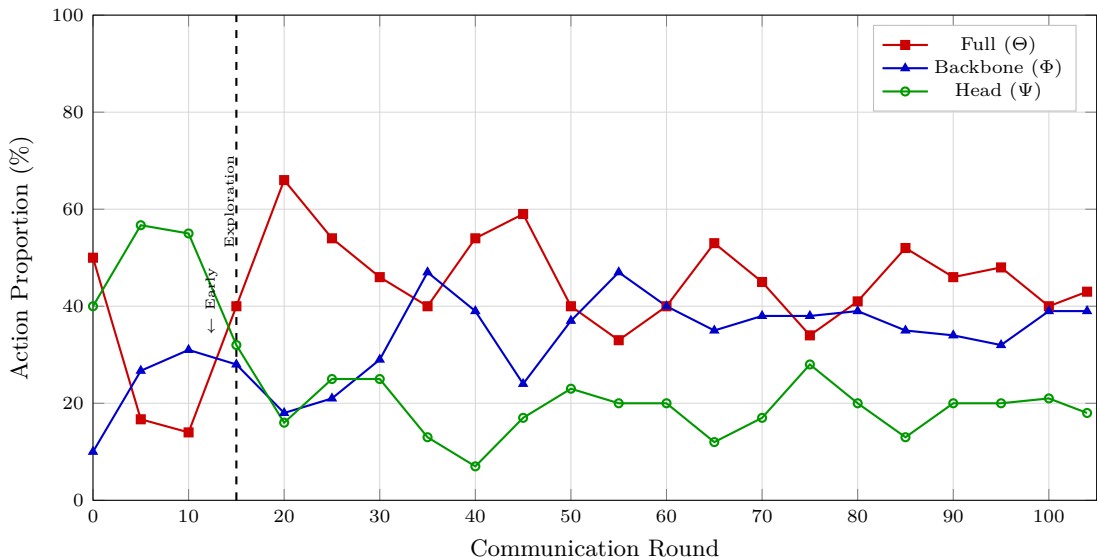

Figure 7: DRL controller action distribution for FedPolicy on CIFAR-10 ($\beta$=0.1).

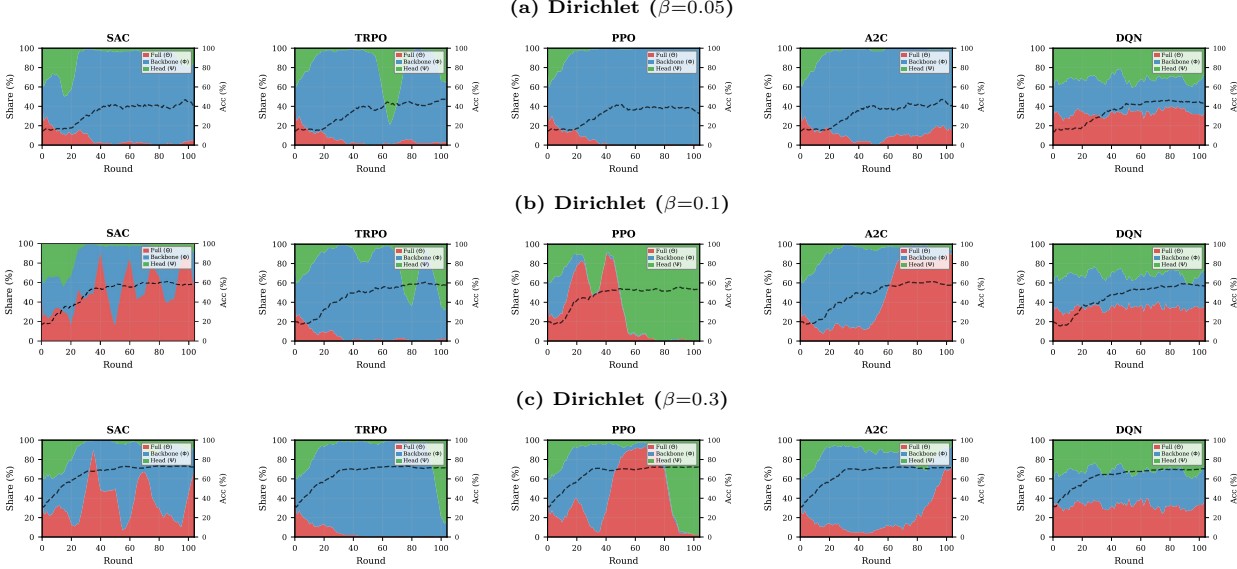

Figure 8: Per-algorithm action evolution (10-round sliding window) for five DRL algorithms across three size-varying Dirichlet label-skew levels. Each panel shows stacked action proportions for Full ($\Theta$), Backbone ($\Phi$), and Head ($\Psi$), together with global test accuracy (dashed line, right axis).

(74.3%). A2C also remains largely backbone-oriented and performs strongly at $\beta$=0.1 and $\beta$=0.3. PPO is the least stable: although it is strongly backbone-dominant at $\beta$=0.05 (93.3%), it shifts to a head-heavy policy at $\beta$=0.1 (65.0% Head), where its accuracy drops to 58.5%.

The effect of policy choice is most visible at $\beta$=0.1. Here, TRPO's backbone-heavy policy (73.4% Backbone) reaches 64.3% accuracy, and A2C's mainly Full/Backbone allocation (49.0%/44.8%) reaches 63.3%, whereas PPO's head-dominant policy performs substantially worse. By contrast, at $\beta$=0.3, the accuracy gap across algorithms narrows to only 1.7 percentage points, indicating that policy specialization matters less when heterogeneity is mild. Overall, the results point to a consistent conclusion: as data heterogeneity becomes

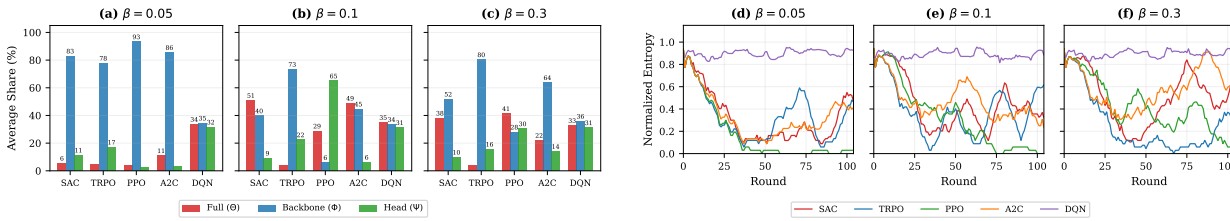

Figure 9: Multi-algorithm policy summary. **Left (a–c):** grouped bar charts of average action proportions per algorithm at each $\beta$. **Right (d–f):** normalized Shannon entropy $H/\log 3$ of action distributions over communication rounds.

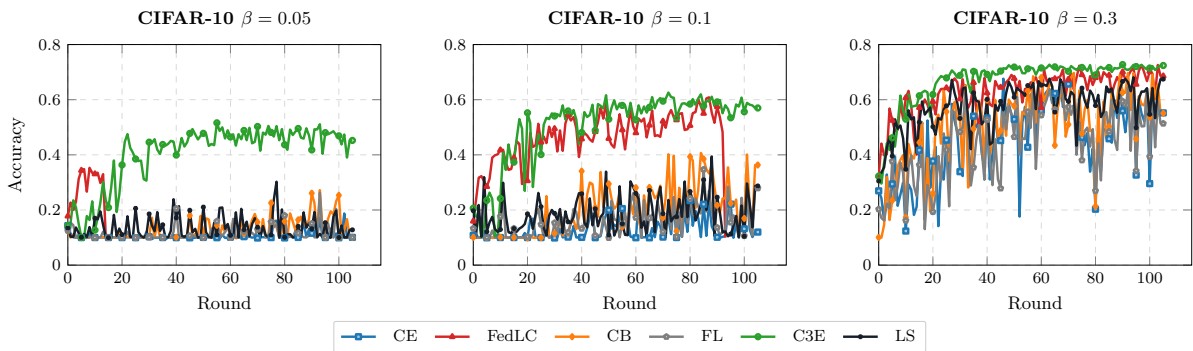

Figure 10: Convergence trajectories of FedPolicy on CIFAR-10 with various local loss functions under $\beta \in \{0.05, 0.1, 0.3\}$.

stronger, effective policies place increasing emphasis on Backbone sharing, suggesting that feature-level transfer is more reliable than uniform full-model synchronization in highly non-IID settings.

## C.4   Ablation Study

Table 10: Effect of local training loss on CIFAR-10 under size-varying Dirichlet label skew. Entries report peak Top-1 accuracy (%) for two client-side training objectives: CE (standard cross-entropy) and C3E (Confusion-Calibrated Cross-Entropy). Best result in each column is shown in **bold**.

| Method | $\beta{=}0.3$ | | $\beta{=}0.1$ | | $\beta{=}0.05$ | |
| | CE | C3E | CE | C3E | CE | C3E |
| --- | --- | --- | --- | --- | --- | --- |
| FedAvg | 64.76 | 71.30 | 31.25 | 49.27 | 16.11 | 39.19 |
| FedProx | 64.82 | 72.10 | 33.79 | 44.20 | 16.31 | 40.12 |
| Ditto | 65.00 | 72.40 | 31.25 | 43.10 | 17.18 | 40.50 |
| CCVR | 67.31 | 73.78 | 38.57 | 51.80 | 15.78 | 48.30 |
| FedSAM | 54.13 | 61.80 | 27.25 | 38.70 | 14.66 | 46.10 |
| FedFA | 63.51 | 71.80 | 28.31 | 41.60 | 17.35 | 46.83 |
| FedNTD | 69.20 | 72.98 | 44.08 | 53.77 | 21.52 | 32.54 |
| FedPart | 66.59 | 72.94 | 23.08 | 53.21 | 13.93 | 22.86 |
| FedCA | 69.28 | 73.46 | 40.68 | 59.01 | 23.83 | 51.62 |
| **FedPolicy** | **71.41** | **74.20** | **48.67** | **63.57** | **28.43** | **55.25** |

Table 11: Hyperparameter sensitivity analysis of the FedPolicy controller under size-varying Dirichlet label skew ($\beta$=0.1). All experiments are evaluated after 55 global communication rounds. Within each parameter block, only the indicated hyperparameter is varied, while the state representation and all remaining controller settings are fixed at the corresponding dataset-specific reference configuration. Entries report the peak global Top-1 accuracy (%). The best-performing configuration within each parameter block is highlighted in bold.

| Controller parameter | Value | CIFAR-10 | CIFAR-100 | ISIC2019 |
|---|---|---|---|---|
| Controller learning rate $\eta_c$ | $1 \times 10^{-4}$ | 49.12 | 31.08 | 56.94 |
| | $5 \times 10^{-4}$ | 50.87 | 32.74 | **58.50** |
| | $1 \times 10^{-3}$ | **55.41** | 33.42 | 56.73 |
| | $5 \times 10^{-3}$ | 52.34 | **35.33** | 57.21 |
| Local reward weight $\lambda_{\text{local}}$ | 0.25 | 50.86 | 31.42 | 56.91 |
| | 0.50 | 53.74 | 33.56 | 57.68 |
| | 0.75 | 51.92 | 32.21 | 56.74 |
| | 1.00 | **55.41** | **35.33** | **58.50** |
| | 1.25 | 49.63 | 30.88 | 56.12 |
| Global reward weight $\lambda_{\text{glob}}$ | 0.25 | 52.18 | 31.27 | 56.48 |
| | 0.50 | 49.94 | 33.41 | **58.50** |
| | 0.75 | 53.07 | **35.33** | 57.14 |
| | 1.00 | **55.41** | 31.86 | 56.66 |
| Hidden width $d_h$ | 64 | 52.36 | 31.18 | 57.02 |
| | 128 | **55.41** | 32.46 | 56.41 |
| | 256 | 49.74 | **35.33** | **58.50** |
| Training interval $I_c$ | 1 | 50.71 | 31.04 | **58.50** |
| | 3 | **55.41** | 32.16 | 57.43 |
| | 5 | 52.84 | **35.33** | 56.51 |
| Update epochs $E_c$ | 3 | 51.24 | 31.72 | **58.50** |
| | 5 | **55.41** | 33.04 | 57.11 |
| | 8 | 49.88 | **35.33** | 56.46 |
| Server validation-subset size $V_s$ | 500 | 49.82 | 31.06 | 56.72 |
| | 1000 | 52.93 | 33.14 | 57.61 |
| | 1500 | **55.41** | 32.28 | 57.08 |
| | 2000 | 54.49 | **35.33** | **58.50** |
| | 2500 | 52.66 | 31.42 | 56.34 |

### C.4.1 Local-Loss Analysis

Figure 10 compares six local loss functions on CIFAR-10 within FedPolicy under three size-varying Dirichlet label-skew levels. The goal of this study is to identify which local objective is most compatible with the proposed adaptive parameter-sharing framework under heterogeneous client distributions.

The results show a clear separation between C3E and the alternative losses, and this difference becomes more pronounced as heterogeneity increases. Under severe skew ($\beta$=0.05), C3E is the only loss that sustains learning in the 55% accuracy range. By contrast, CE, CB, FL, and LS remain close to chance level for most of training, while FedLC improves only briefly before deteriorating. At $\beta$=0.1, C3E again maintains the strongest and most stable convergence, whereas the remaining losses either saturate early or exhibit unstable trajectories. Even at the mildest setting ($\beta$=0.3), where the gap is smaller, C3E still achieves the best peak accuracy with the smoothest training behavior.

Table 10 shows that replacing CE with C3E consistently strengthens all compared methods across all heterogeneity levels. However, the advantage of FedPolicy cannot be explained by the local loss alone. Even when the baselines are equipped with the same C3E objective, FedPolicy remains the top-performing method at every $\beta \in \{0.3, 0.1, 0.05\}$. In this appendix table, CCVR is the strongest C3E baseline at $\beta$=0.3, while

FedCA is strongest at $\beta$=0.1 and $\beta$=0.05, reaching 73.78%, 59.01%, and 51.62%, respectively. FedPolicy reaches 74.20%, 63.57%, and 55.25%, corresponding to gains of +0.42, +4.56, and +3.63 percentage points. Under the standard CE setting, FedCA is likewise the strongest baseline at all three heterogeneity levels; FedPolicy improves over it by +2.13, +7.99, and +4.60 percentage points. FedPart is included under the same CE and C3E local-loss settings for this comparison. These results indicate that the benefit of FedPolicy is additive to that of stronger local optimization and becomes especially important under severe skew.

### C.4.2 Controller Hyperparameter Sensitivity

We examine whether the learned redistribution policy depends on a narrowly selected controller configuration. Table 11 reports a one-factor-at-a-time sensitivity analysis over the controller learning rate $\eta_c$, local reward weight $\lambda_{\mathrm{local}}$, global reward weight $\lambda_{\mathrm{glob}}$, hidden width $d_h$, controller-training interval $I_c$, controller update epochs $E_c$, and server validation-subset size. Within each parameter block, only the indicated hyperparameter is varied, while the state representation and all remaining controller settings are fixed at the corresponding dataset-specific reference configuration. All experiments use size-varying Dirichlet label skew ($\beta$=0.1), 100 clients, 5 participating clients per round, and 55 global communication rounds.

Table 11 shows that FedPolicy remains effective across a range of controller settings while still benefiting from dataset-specific tuning. For CIFAR-10, the strongest settings are close to the reference configuration, including $\eta_c$=$10^{-3}$, $\lambda_{\mathrm{local}}$=1.0, $\lambda_{\mathrm{glob}}$=1.0, $d_h$=128, $I_c$=3, and $E_c$=5. CIFAR-100 and ISIC2019 favor somewhat different controller schedules and model widths, indicating that the best controller configuration can depend on dataset complexity and the validation signal available to the policy. The reward-weight blocks show that the local reward term is important across all three datasets, with $\lambda_{\mathrm{local}}$=1.0 giving the best result in each case, while the preferred global reward weight is dataset-dependent. The validation-subset block shows that a moderate held-out subset is sufficient: CIFAR-10 peaks at $V_s$=1500, whereas CIFAR-100 and ISIC2019 peak at $V_s$=2000, and increasing the subset to 2500 does not improve performance.

### C.4.3 Client-Dropout Robustness

We evaluate robustness to synchronous client dropout on CIFAR-10 with size-varying Dirichlet label skew ($\beta$=0.1), 100 clients, 10 candidate clients sampled per round, ResNet-18, five local epochs, and 105 communication rounds. After the candidate set is sampled, each selected client is independently removed with probability $p_{\mathrm{drop}} \in \{0.1, 0.3, 0.5\}$ before local training and aggregation. Thus, $p_{\mathrm{drop}}$ denotes the per-selected-client failure probability, not the fraction of all 100 clients removed permanently. With 10 candidates per round, the corresponding expected numbers of participating clients are 9, 7, and 5, respectively. If all candidates are removed, one is retained to preserve a valid synchronous round. The same data partition, candidate-client sampling, seed, and training budget are used for FedAvg, FedProx, FedNTD, FedSAM, FedCA, and FedPolicy. Each method retains its configured local objective and aggregation mechanism. The FedPolicy entry uses the same SAC redistribution controller profile as the corresponding robustness experiment.

Table 12 shows that FedPolicy remains effective across the evaluated dropout probabilities and outperforms all compared methods in each column. Its peak accuracy stays within a 2.17-point range, from 51.63% to 53.80%, and exceeds the strongest non-FedPolicy baseline, FedCA, by 6.96, 7.06, and 7.42 percentage points at $p_{\mathrm{drop}}$=0.1, 0.3, and 0.5, respectively. The completed baseline set, including Ditto, CCVR, FedFA, and FedPart, follows the same dropout protocol and remains below FedCA and FedPolicy in this evaluation. The non-monotonic peak values should not be interpreted as evidence that a particular dropout rate improves learning. Under heterogeneous data, changing the number and composition of surviving clients changes the update trajectory, and the maximum observed over 105 rounds is sensitive to transient fluctuations. Because each configuration uses one seed, the study supports resistance to performance collapse under the evaluated random-dropout protocol, but not a monotonic relationship between dropout and accuracy.

### C.4.4 Aggregation-Compatibility and Selective-Initialization Stability

Selective redistribution mixes a newly aggregated block with a retained local block, so the retained component may differ from the current global model. We do not interpret this retained component as a stale asynchronous update. Rather, the retained block is an intentional initialization choice: for example, when

Table 12: Synchronous client-dropout robustness on CIFAR-10 under size-varying Dirichlet label skew ($\beta$=0.1). In each round, 10 candidate clients are first sampled, and each candidate is independently removed before local training with probability $p_{\text{drop}}$. Entries report peak global Top-1 accuracy (%) over 105 communication rounds. Best result in each column is shown in bold.

| Method | $p_{\text{drop}}$=0.1 | $p_{\text{drop}}$=0.3 | $p_{\text{drop}}$=0.5 |
|---|---|---|---|
| FedAvg | 38.50 | 38.16 | 32.18 |
| FedProx | 38.91 | 38.01 | 30.74 |
| Ditto | 39.84 | 38.47 | 35.62 |
| FedNTD | 36.63 | 38.91 | 34.98 |
| FedSAM | 36.38 | 34.18 | 33.14 |
| CCVR | 43.78 | 42.16 | 39.71 |
| FedFA | 44.29 | 42.83 | 40.36 |
| FedPart | 42.51 | 40.72 | 38.43 |
| FedCA | 46.84 | 44.57 | 44.50 |
| **FedPolicy** | **53.80** | **51.63** | **51.92** |

Table 13: Aggregation-compatibility evaluation on CIFAR-10 under size-varying Dirichlet label skew ($\beta$=0.1) using C3E as the local training loss. For each aggregation method, the baseline column uses uniform Full redistribution, while the FedPolicy column uses selective post-aggregation redistribution with the same aggregation method. Entries report peak global Top-1 accuracy (%) over 105 communication rounds, and Gain is reported in percentage points (p.p.).

| Aggregation Method | Full redistribution (Baseline) | Selective redistribution (FedPolicy) | Gain (p.p.) |
|---|---|---|---|
| FedCA | 59.01 | **61.87** | +2.86 |
| Mean | 56.84 | **62.84** | +6.00 |
| FedAvg | 49.27 | **56.68** | +7.41 |
| FedAvgM | 48.41 | **55.89** | +7.48 |
| FedAdam | 58.46 | **63.25** | +4.79 |
| Elastic | 41.99 | **52.51** | +10.52 |

a client receives only the global head, its local backbone is preserved so that additional local optimization can continue adapting representation parameters to that client's local distribution, while the classifier head receives global correction. The relevant question is therefore whether the resulting selective initialization remains compatible with server aggregation.

To examine this point, Table 13 reports a matched aggregation-compatibility evaluation on CIFAR-10 with size-varying Dirichlet label skew ($\beta$=0.1), 100 clients, 10 participating clients per round, ResNet-18, C3E as the local training loss, and 105 communication rounds. We compare uniform Full redistribution and FedPolicy selective redistribution under the same aggregation operators, including FedCA.

The results show no evidence of collapse from combining retained and newly aggregated blocks in this evaluation. Selective redistribution improves peak global Top-1 accuracy under all six matched aggregation settings, with gains ranging from +2.86 to +10.52 percentage points. We therefore treat block mismatch as a quantity that should be monitored and controlled, rather than as evidence that retained local blocks are inherently destabilizing.

### C.5 Cost Analysis

For practical deployment, the adaptive controller should improve performance without introducing a meaningful systems burden. We therefore evaluate FedPolicy along three complementary dimensions: per-round computational overhead, communication cost, and the joint cost–accuracy trade-off.

Table 14: Per-round wall-clock time decomposition (seconds) on CIFAR-10 ($\beta$=0.1). The — indicates that baseline costs do not include controller overhead.

| Method | Local Train (s) | Agg. (s) | DRL (s) | Total (s) |
|---|---|---|---|---|
| FedAvg | 20.90 | 0.012 | — | 20.91 |
| FedProx | 20.90 | 0.012 | — | 20.91 |
| FedNTD | 20.90 | 0.012 | — | 20.91 |
| Ditto | 20.90 | 0.012 | — | 20.91 |
| FedSAM | 19.44 | 0.026 | — | 19.47 |
| FedFA | 27.44 | 0.030 | — | 27.47 |
| CCVR | 31.06 | 0.023 | — | 31.08 |
| FedCA | 20.92 | 0.087 | — | 21.79 |
| **FedPolicy** | 20.92 | 0.011 | 0.021 | 20.952 |

### C.5.1 Computational Overhead

Table 14 reports the per-round wall-clock time on CIFAR-10 under $\beta$=0.1, decomposed into local training, server aggregation, and controller cost. The additional cost of the DRL controller is only 0.021 s per round, which amounts to approximately 0.10% of the total runtime and about 2.2 s over the full 105-round training budget. As a result, the overall per-round time of FedPolicy is 20.95 s, essentially matching FedAvg-style training (20.91 s) and remaining well below more expensive baselines such as FedCA (21.79 s), FedFA (27.47 s), and CCVR (31.08 s). This indicates that the performance gain of FedPolicy is not achieved at the expense of noticeable computational overhead.

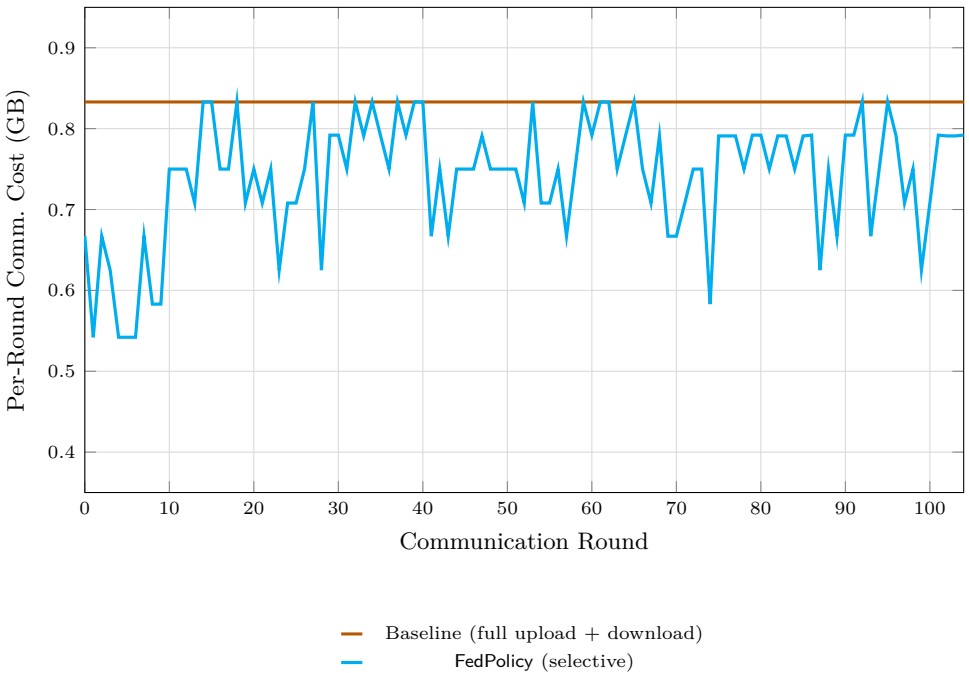

Figure 11: CIFAR-10 ($\beta$=0.1): per-round communication cost. Baselines incur a constant 0.833 GB/round because each selected client uploads and downloads the full model in every round; FedPolicy remains lower in early rounds and approaches baseline as Full updates become more frequent.

### C.5.2 Communication Efficiency

We next examine communication accounting under selective broadcast. In baseline methods, each of the $C_r$=10 selected clients uploads and downloads the full model in every round, yielding a total bidirectional cost of

$$C_r \times 2 \times |\Theta| = 10 \times 2 \times 42.66 \text{ MB} \approx 0.833 \text{ GB/round.}$$

Over $T$=105 rounds, this corresponds to approximately $87.5$ GB of total communication. In FedPolicy, uploads remain full-model, while downloads are selective (Full/Backbone/Head). Therefore, per-round savings come from reduced downlink payload, and cumulative savings additionally reflect fewer rounds needed to reach a target accuracy.

Figure 11 shows that FedPolicy remains below the baseline cost in the early stage of training, where Head and Backbone updates are selected more frequently, and then gradually approaches the full-model baseline as Full updates become more common. In practice, the per-round cost of FedPolicy starts at roughly 0.54–0.67 GB and increases later in training as the policy shifts toward broader synchronization. This behavior is consistent with the learned action dynamics: communication is reduced precisely in the regime where selective sharing is used most aggressively.

## D Discussion

### D.1 Key Observations

- The main contribution of FedPolicy lies in improving the match between global redistribution and client-specific learning conditions. Its advantage is modest when heterogeneity is mild, but becomes increasingly pronounced as client distributions grow more skewed, where uniform rebroadcast becomes less appropriate.

- The learned policy behavior is consistent with this trend. As heterogeneity increases, the controller places progressively greater emphasis on Backbone sharing, suggesting that feature-level transfer remains broadly useful across clients, whereas classifier-level parameters are more tightly coupled to local label structure. The ablations support the same interpretation: parameter divergence is the most informative state signal, and combining local and global reward terms yields the most stable behavior.

- The controller-ablation and local-training-loss comparison isolate post-aggregation redistribution as the active design choice in FedPolicy. The proposed method keeps server aggregation fixed to the mean rule and learns only the post-aggregation sharing decision. This comparison includes both standard CE and C3E settings, showing that the gain arises from the proposed adaptive post-aggregation sharing mechanism in addition to stronger local optimization.

- The comparison with static and Random sharing rules provides empirical evidence that the improvement in stronger heterogeneous regimes is associated with state-dependent learned redistribution rather than a single fixed preference for Full, Backbone, or Head updates. No single static heuristic performs best across all regimes. This evidence does not imply policy optimality or guarantee that each selected action minimizes redistribution mismatch.

- The performance gain is achieved with negligible practical overhead. The controller adds only a very small fraction to the total per-round runtime, while sharing selective parameter blocks to clients reduces communication cost, leading to a more favorable cost-to-accuracy trade-off.

### D.2 Privacy Considerations

FedPolicy does not require raw data sharing, but the method does not by itself provide a formal privacy guarantee beyond the standard federated setting. The controller operates on model updates together with auxiliary client-side statistics such as soft confusion information and local accuracy. Although these signals are compact, they are still derived from private local data and may reveal information about class composition

or client-specific difficulty. The present results therefore establish utility, not formal privacy protection. When formal privacy guarantees are required, FedPolicy can be paired with standard privacy-preserving techniques such as differential privacy, clipping with calibrated noise, secure aggregation, and explicit privacy accounting.

### D.3 Limitations

The present evaluation is limited to image classification, with LeNet-5 on FMNIST and a backbone—head decomposition based primarily on ResNet-18 for CIFAR experiments. While the general idea is not tied to a specific architecture, its usefulness depends on whether a meaningful separation exists between transferable representation parameters and more client-specific prediction layers. The current action space is also deliberately coarse, consisting only of Full, Backbone, and Head updates. This makes the controller simple and interpretable, but it also limits the granularity of adaptation. Evaluating whether the same redistribution principle remains effective for other modalities and architectures, such as transformer-based models, encoder–decoder networks, adapter-based models, or architectures without an explicit backbone/head split, is an important direction for future work. Similarly, the present state and reward design is tailored to classification through confusion statistics and validation accuracy; applying learned selective redistribution to other learning problems would require problem-specific RL formulations with appropriate state descriptors, actions, and reward signals.

