# OpenReview forum: "FedPolicy: An RL-Guided Redistribution Policy for Synergizing Local-Global Optimization in Federated Learning"
_TMLR — Under review for TMLR_

### Review · Reviewer_ho3Y · 2026-05-19

**Summary Of Contributions:**

Tha authors consider a RL guided way to perform server aggregation of models. The core idea is that after aggregation, the model should not always be broadcast uniformly to every client. Instead the controller decides to either send a) the full model b) only the backbone/feature extractor or c) only the classifier head.

**Additional Comments:**

This is a promising and well-motivated FL paper with a clean central idea, I am happy to accept if the authors answer my minor concerns.

**Audience:**

Yes

**Audience Explanation:**

Using an RL guided framework in FL is not new. But deciding the complexity of the model is fairly new, as far as I know.

**Claims And Evidence:**

Yes

**Claims Explanation:**

There is good support for the results claimed.

**Requested Changes:**

The RL formulation may be heavier than necessary. The action space is only three choices, and some static baselines are already competitive. Why can't whether a simpler contextual bandit, supervised policy, or rule-based controller be used instead?

The theoretical analysis is somewhat disconnected from the learned policy. The paper explicitly says the theory analyzes selective redistribution after an action is chosen, not convergence of the RL controller. That is reasonable, but it weakens the theoretical support for the full FedPolicy algorithm.

---

### Review · Reviewer_mEwC · 2026-06-01

**Summary Of Contributions:**

This paper studies federated learning under heterogeneous local data distributions and asks whether the post-aggregation redistribution step should remain uniform across clients. The proposed method, FedPolicy, uses a DRL controller to decide whether each participating client should receive the full global model, only the backbone, or only the head at each communication round. The method is evaluated on FMNIST, CIFAR-10, and CIFAR-100, and the paper also includes a theoretical section and several ablations.

While the high-level question is interesting, I do not find the current submission mature enough for publication: the motivation is not fully explained, the theory section is non-informative, and the empirical section does not cleanly isolate what is gained over simpler alternatives.

**Additional Comments:**

- The claim in Section 2.4.2 that large parameter mismatch favors selective redistribution over full overwrite is counterintuitive and not experimentally validated. A client that has diverged substantially from the global model may be sending noisy updates that harm aggregation, so the opposite conclusion could also hold. An ablation specifically addressing this would strengthen the paper.
- The method currently assumes a backbone/head decomposition that is architecture-specific. The paper should discuss how the approach generalizes to architectures without a clear such separation.
- The paper should clarify whether the method is restricted to classification tasks, given that the confusion matrix state representation and validation accuracy reward are both classification-specific.
- Compatibility with aggregation rules other than FedAvg is not demonstrated. Showing that FedPolicy can be combined with e.g. FedCA aggregation would help establish that it targets a genuinely orthogonal mechanism. The interplay with C3E loss is also not very well established.

**Audience:**

Yes

**Audience Explanation:**

The problem formulation is novel and relevant. Post-aggregation redistribution is an underexplored axis in FL, and framing it as a sequential decision problem is a reasonable and interesting direction. If the empirical evidence were stronger and the theoretical claims informative, this could be a useful contribution. The backbone/head asymmetry motivation is also practically grounded.

**Claims And Evidence:**

No

**Claims Explanation:**

The theoretical analysis does not support the claims:

- Lemma 1 is a direct application of the definition of L-smoothness to a specific point. Calling it a lemma is misleading.
- Theorem 1 shows that if a better action exists, it yields lower loss. This is essentially tautological and provides no guarantee that the policy finds such an action.
- Theorem 3 is the most serious issue: it assumes that selective redistribution induces lower drift than uniform overwrite ($\Gamma^2_{Pol} \le \rho \Gamma^2_{Full}$), which is precisely what should be proved. As stated, it gives no theoretical guarantee that FedPolicy achieves lower drift than FedAvg, and the theoretical section should either be substantially reworked or removed.
- The central issue for not aggregating uniformly is that it introduces much larger drift between clients, which can destabilize training. The paper does not provide any theoretical reason why this drift would not be harmful for convergence, while implying the reverse.

The empirical evidence does not entirely support the central claims either:

- Results are reported only on toy datasets (FMNIST, CIFAR-10, CIFAR-100). More realistic FL benchmarks such as FLamby would be needed to support broad claims about heterogeneous federated learning.
- The main results rely heavily on C3E, a metric introduced in a very recent paper (FedCA), and is never defined in the paper. Readers cannot interpret the results independently and the FedCA paper is behind paywall.
- The DQN controller learns a near-uniform policy (≈33%/35%/32% across all β) yet remains competitive with specialized controllers. This directly undermines the claim that learning the redistribution policy is the source of gains.
- Table 5 reports Full-Only performance that is substantially inconsistent with FedAvg results in Table 2. Since Full-Only should recover FedAvg, this discrepancy is unexplained and raises concerns about experimental consistency. I guess this is due to C3E local loss but this undermines the claim that the method is evaluated outside of any other changes. All other baselines could be also tuned with this metric otherwise.
- Some results lack standard deviations in most tables, and the number of repetitions is not stated in Table 2.
- The comparison with more recent SOTA methods on CIFAR-10 with similar settings suggests the reported baseline numbers may be below current best results, making the improvements harder to contextualize. See for instance for results on CIFAR10 with beta=0.3 here: https://www.wizwand.com/sota/federated-learning-on-cifar-10-100-clients-dirichlet-0-3. Note that inclusion rate is 2% so 500 rounds is comparable with the 10% inclusion rate with 100 rounds in the paper.

**Requested Changes:**

- Theoretical section: Either provide a proof that FedPolicy actually induces lower drift than uniform redistribution, or remove/reframe the theoretical claims to reflect what is and is not proved. In its current form, Theorem 3 assumes its own conclusion and is misleading.
- Experiments: Add evaluation on at least one realistic FL benchmark (e.g. FLamby). Report standard deviations and number of seeds consistently across all tables and figures, not just in the first table.
- C3E: Define it explicitly in the paper, or replace it with a standard metric. Relying on a self-cited concurrent paper without definition is insufficient.
- Table 5 vs Table 2 discrepancy: Explain or fix the inconsistency between Full-Only and FedAvg results.
- DQN analysis: Discuss explicitly why a near-uniform policy performs competitively, and what this implies about the necessity of policy learning.
- Figure 5: Add the missing accuracy panel.
- Section 2.1: Formally introduce the standard FL optimization problem before presenting the method.
- Section 2.6: Justify the use of t-1 instead of t in the reward definition.
- Contributions: Merge the contributions paragraph and bullet points. This is currently redundant.
- Figure 2: Increase font size and simplify; key elements (MUX, reward computation) are not self-explanatory.
- Table 3: it is unclear if the models are here trained with C3E or if C3E is the metric that is reported.

---

### Review · Reviewer_xzBx · 2026-06-17

**Summary Of Contributions:**

The paper’s main contribution is to identify post-aggregation redistribution as an underexplored design choice in federated learning and to propose FedPolicy, an RL-guided mechanism that adaptively decides which part of the global model should be sent back to each client. This is an interesting perspective because it separates redistribution from local optimization and aggregation, and the experiments suggest that selective sharing can improve accuracy, convergence, and communication efficiency under heterogeneous data. A key strength is the clear motivation and broad empirical comparison across datasets and heterogeneity levels. A potential weakness is that the method is evaluated mainly on image classification with a relatively coarse backbone/head action space, so its generality to other architectures, tasks, and more realistic federated settings remains less clear.

**Additional Comments:**

No.

**Audience:**

Yes

**Audience Explanation:**

Yes. At least some members of the TMLR audience would likely be interested in the findings, particularly researchers working on federated learning, heterogeneous optimization, personalization, and adaptive training methods. The paper studies an underexplored stage of the FL pipeline—post-aggregation redistribution—and shows that client-specific selective sharing can improve accuracy, convergence, and communication efficiency under non-IID data. While the scope is somewhat specialized and mainly evaluated on image classification, the core idea is relevant to a recognizable audience within machine learning.

**Broader Impact Concerns:**

No.

**Claims And Evidence:**

Yes

**Claims Explanation:**

The paper provides extensive experiments across multiple datasets and heterogeneity levels, showing consistent accuracy gains, faster convergence, and favorable communication/runtime trade-offs. The ablation studies further strengthen the evidence by comparing learned policies with static sharing rules and by isolating the effects of state and reward design. However, the theoretical analysis is somewhat limited because it abstracts away the RL controller’s learning dynamics, and the empirical evaluation is mostly restricted to image classification with a coarse backbone/head action space. Thus, the evidence is convincing for the tested settings, but broader generalization remains insufficiently established.

**Requested Changes:**

**Critical to acceptance:**

1. **Broaden the empirical validation beyond the current image-classification setting.** The paper’s main claim is about heterogeneous federated learning in general, but the experiments are mainly on FMNIST/CIFAR with LeNet/ResNet-style backbone-head decompositions. Additional evidence on another architecture, task type, or more realistic FL scenario would make the generality of the claim much more convincing.

2. **Clarify the novelty boundary with existing partial-sharing, personalization, and RL-based FL methods.** The paper positions post-aggregation redistribution as underexplored, but related ideas such as personalized layers, partial network updates, and RL-guided FL coordination are close enough that the distinction should be made sharper.

3. **Strengthen the explanation of why the learned RL policy is necessary.** The static-policy ablations are useful, but the paper should more clearly explain when the learned policy outperforms simple heuristics and whether the improvement justifies the added algorithmic complexity.

**Would strengthen the work:**

4. **Improve the theoretical section’s framing.** The current theory supports selective initialization after an action is chosen, but it does not analyze the learned RL controller itself. The paper should state this limitation more explicitly and avoid implying a full convergence theory for FedPolicy.

5. **Add robustness experiments under more practical FL conditions.** Client dropout, asynchronous participation, system heterogeneity, and time-varying client distributions would make the practical relevance stronger.

6. **Provide more policy interpretability.** The action-distribution plots are helpful, but the paper could better connect specific client states, such as parameter divergence or confusion patterns, to the selected redistribution action.

7. **Report sensitivity to controller hyperparameters and validation-set size.** Since the controller uses server-side validation accuracy and RL training, it would be useful to know how stable the method is under different controller settings and smaller held-out validation splits.

8. **Clarify privacy implications of sharing soft confusion statistics.** The discussion notes that these statistics may leak information, but a more concrete discussion or mitigation strategy would strengthen the paper.